# Spreading ridge migration enabled by plume-ridge de-anchoring

Ben Mather [1] ✉, Maria Seton [1], Simon Williams [2], Joanne Whittaker [2], Rebecca Carey [3], Maëlis Arnould[4], Nicolas Coltice[5] & Robert Duncan[6]

It has long been recognised that spreading ridges are kept in place by competing subduction forces that drive plate motions. Asymmetric strain rates pull spreading ridges in the direction of the strongest slab pull force, which partially explains why spreading ridges can migrate vast distances. However, the interaction between mantle plumes and spreading ridges plays a relatively unknown role on the evolution of plate boundaries. Using a numerical model of mantle convection, we show that plumes with high buoyancy flux (>3000 kg/s) can capture spreading ridges within a 1000 km radius and anchor them in place. Exceptionally high buoyancy fluxes may fragment the overriding plate into smaller plates to accommodate more efficient plate motion. If the plume buoyancy flux wanes below 1000 kg/s the ridge may be de-anchored, leading to rapid ridge migration rates when combined with asymmetric plate boundary forces. Our results show that plume-ridge de-anchoring may have contributed to the rapid migration of the SE Indian Ridge from 43 million years ago (Ma) due to waning buoyancy flux from the Kerguelen plume, supported by magma flux estimates and radiogenic isotope geochemistry of eruption products. The plume-ridge de-anchoring mechanism we have identified has global implications for the evolution of plate boundaries near mantle plumes.

Plate tectonic motions are primarily driven by slab pull at subduction zones[1] and, to a lesser extent, plume push at spreading ridges[2]. The migration of spreading ridges is thought to be driven by asymmetric plate motions on either ridge flank transmitted from far-field slab pull forces[3,4]. Within this canonical model of plate motion, the geometry and location of spreading ridges are independent of lower mantle dynamics. However, this fails to explain observed interactions between spreading ridges and mantle plumes[5], such as observed instances of ridge capture and successive ridge jumps towards a plume[6], how the proximity of ridges to plumes may be accommodated by extensive transform faults[7], and why ridge migration rates are perturbed in the vicinity of mantle plumes[8,9]. Moreover, while numerical models predict that mantle plumes and spreading centres should preferentially align[10], the dynamics and temporal evolution of plume–ridge interaction and its influence on the geometry and migration of spreading ridges is not fully understood.

In nature, plume–ridge interactions are commonly preserved as conjugate large igneous provinces (LIPs)[11], which form on each flank of a spreading ridge as new crust is created[5]. In most cases, the de-anchoring of plumes and spreading ridges is demarcated by a switch from conjugate LIP formation or ridge formation, via a plume head beneath a spreading ridge, to trails of intraplate hotspot volcanoes on

[1]EarthByte Group, School of Geosciences, The University of Sydney, Sydney, NSW, Australia. [2]Institute of Marine and Antarctic Studies, University of Tasmania, Hobart, TAS, Australia. [3]CODES School of Natural Sciences, University of Tasmania, Hobart, TAS, Australia. [4]Laboratoire de Géologie de Lyon – Terre, Planètes, Environnement, LGL-TPE, University of Lyon, UCBL, ENSL, UJM, CNRS, Villeurbanne, France. [5]Observatoire de la Côte d'Azur, Université Côte d'Azur, CNRS, IRD, Géoazur, Valbonne, France. [6]College of Earth, Ocean, and Atmospheric Sciences, Oregon State University, Corvallis, OR, USA. ✉e-mail: ben.mather@sydney.edu.au

one plate signifying the plume and spreading ridge have separated. For example, the Kerguelen plume and SE Indian Ridge record three stages of interaction: (i) conjugate LIP formation while seafloor spreading was slow (>80 Ma), (ii) the SE Indian Ridge remained coupled to the Kerguelen Plume as India rapidly migrated northward, due to strong slab pull forces[12] and/or ridge-push from the Reunion plume[2], forming the Ninety-East Ridge on the Indian plate (80–43 Ma)[13], and (iii) de-anchoring at 43 Ma as the SE Indian Ridge migrated northwards to accommodate the rapid motion of the Australian plate, which resulted in a switch to low-volume intraplate volcanism on the Kerguelen Plateau[14]. De-anchoring has been preserved in the geometry of the SE Indian Ridge by spreading segments that are offset by transform faults towards the Kerguelen plume beneath the Antarctic Plate[7] and the magma-poor William's Ridge–Broken Ridge rifted margins[15]. In conjunction with insights into plume–ridge interactions from a numerical model of whole-mantle convection, we analyse the geochemistry and magma flux along the Ninety-East Ridge as a key example from nature to understand the drivers of plume–ridge de-anchoring and its feedback with the geometry of plate boundaries and ridge migration rates.

## Results

### Ridge capture

To understand the processes responsible for ridge de-anchoring and ridge migration rates, it is first necessary to understand the dynamics of ridge capture by mantle plumes. We adopt a 3D spherical numerical model of mantle convection at Earth-like vigour[16]. The model produces self-consistent plate-like behaviour and dynamic mantle plumes that reproduce plume excess temperatures, and buoyancy fluxes statistically comparable to Earth observations[16,17]. From the long model timespan of 323 Myr, we track the evolution of multiple mantle plumes and spreading ridges through time. Plume and spreading ridge dynamics are not prescribed but naturally emerge from solving the set of equations, which enables us to study feedbacks in constantly evolving plume–ridge interactions.

The model resolves 40 instances of plume–ridge interactions where ~50% of all model plumes come within 1000 km of a ridge during the 323 Myr timespan. Of those plumes that approach ridges, 13 out of 40 exhibit plume–ridge capture and de-anchoring dynamics. Box plots of the plume buoyancy flux (see "Methods" section), calculated across these 13 instances, show that high flux generally corresponds to ridge capture and low flux corresponds to ridge de-anchoring (Fig. 1). We have analysed the dynamics of all 13 plume–ridge coupled systems and present one representative example of plume–ridge interaction where the ~1000 km proximity of a relatively stationary plume conduit to a spreading ridge causes the ridge to be captured (Fig. 2a–c and Supplementary Movie 1). Ridges appear in the model where the tensile forces are stronger than the strength of the lithosphere, which causes the viscosity to drop and localises deformation. Here, the plume is located ~1000 km from the ridge for nearly 30 Myr in the leadup to the ridge capture. Plume–ridge attraction is evidenced by extensive transform-like boundaries, which displace ridge segments progressively closer to the plume and a 10° tilting of the plume towards the ridge (Fig. 2a). The tilting of the plume in concert with the offset of ridge segments closer to the plume demonstrates the attraction is mutual, which is an advance on previous numerical simulations of plume–ridge interactions, particularly 2D simulations where transform-like offsets are, by definition, impossible.

In the 30 Myr leadup to ridge capture, the mean plume buoyancy flux, averaged from the core-mantle boundary to the surface, increases from 4500 kg/s to 6000 kg/s and thermally weakens the overriding plate (Fig. 3a). By thinning this boundary layer, the plume makes the boundary weaker and helps to localise a ridge segment if the configuration of forces is favourable for plate fragmentation. This is manifested in a new proto-ridge extending from above the plume to the original spreading ridge (Fig. 2b). The previous spreading ridge

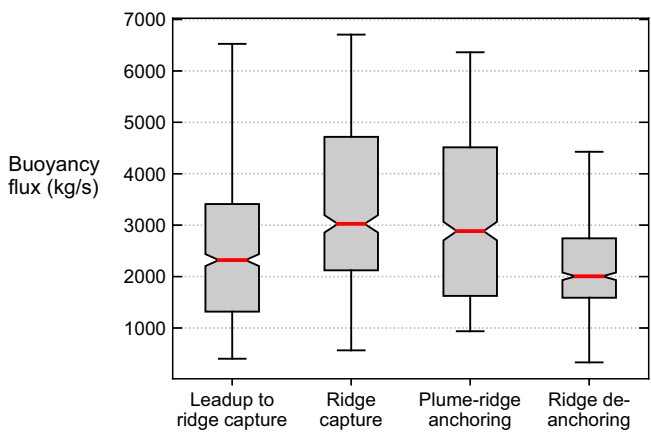

**Fig. 1 | Box plots of plume buoyancy flux for different stages of plume–ridge interaction.** Box plots are calculated from 13 instances of plume–ridge interaction in the numerical model where the median is indicated by red lines and the interquartile range (Q1–Q3) is highlighted in grey. The plume buoyancy flux is calculated from the product of the mean cross-sectional area of the plume, excess mantle temperature, the upwelling velocity, the mantle reference density and reference thermal expansivity (refer to "Methods" section). The buoyancy flux of the plume in the leadup to ridge capture is calculated from timesteps where the ridge and plume are within 1000 km of each other prior to plume–ridge capture; the flux at ridge capture is calculated from 20 timesteps overlapping the point where the distance between the plume and ridge approach zero; the flux at plume–ridge anchoring is calculated at all timesteps between ridge capture and ridge de-anchoring; the flux at ridge de-anchoring is calculated from 20 timesteps overlapping the point where the distance between the plume and ridge begins to increase. The leadup to ridge capture is characterised by moderate plume buoyancy fluxes (median of 2300 kg/s) and a 1400–3400 kg/s interquartile range; plumes reach their zenith buoyancy flux during ridge capture (between 2100 and 4750 kg/s), and their widest interquartile range during plume–ridge anchoring (between 1750 and 4500 kg/s); the lowest plume buoyancy flux occurs during plume–ridge de-anchoring (median of 2000 kg/s) and narrow interquartile range of 1650–2750 kg/s.

develops into a triple junction directly on top of the plume as spreading along the proto-ridge establishes a new plate boundary (Fig. 2c). Following the ridge capture, the triple junction is anchored to the reference frame of the plume and the spreading rate increases from 5.5 to 8.5 ± 1.5 cm/yr (measured by the velocity of the plates relative to the plume, Fig. 3c).

We find that plumes of higher buoyancy flux locally increase the spreading rate along ridge segments (Fig. 4a), which is largely because the plume is strong enough to break the plate into smaller pieces (like a triple junction) to accommodate more efficient plate motion. Now anchored to the triple junction, the high plume buoyancy flux maintains a weak zone in the boundary layer, which remains while the configuration of plate boundary forces is consistent. In this example, the plume and triple junction stay coupled for the remaining 60 Myr of the time series while the plume flux remains high, and the plume conduit straightens sub-vertically in the upper mantle.

2D dynamic models have shown that an increased rate of magmatic heating, associated with higher plume buoyancy flux, is required to initiate a ridge jump under faster spreading[6]. In agreement with the 2D models, the increased buoyancy flux in our 3D spherical simulation, combined with the absence of any significant variation in seafloor spreading rates in the first 30 Myr of the simulation, causes the ridge to jump 1000 km on top of the plume conduit (Fig. 3b) where the plume has thermally weakened the overriding plate (Supplementary Fig. 2b). The timescale of the ridge jump occurs within 1 Myr, consistent with the $10^5$–$10^6$ year time frame seen in 2D models[6]. At the time of the ridge jump, the plume buoyancy flux reaches 6700 kg/s at its peak (Fig. 3a) which is within the range of the present-day Hawaiian plume (2800–8700 kg/s[18,19]). Spreading ridges approaching within 1000 km of

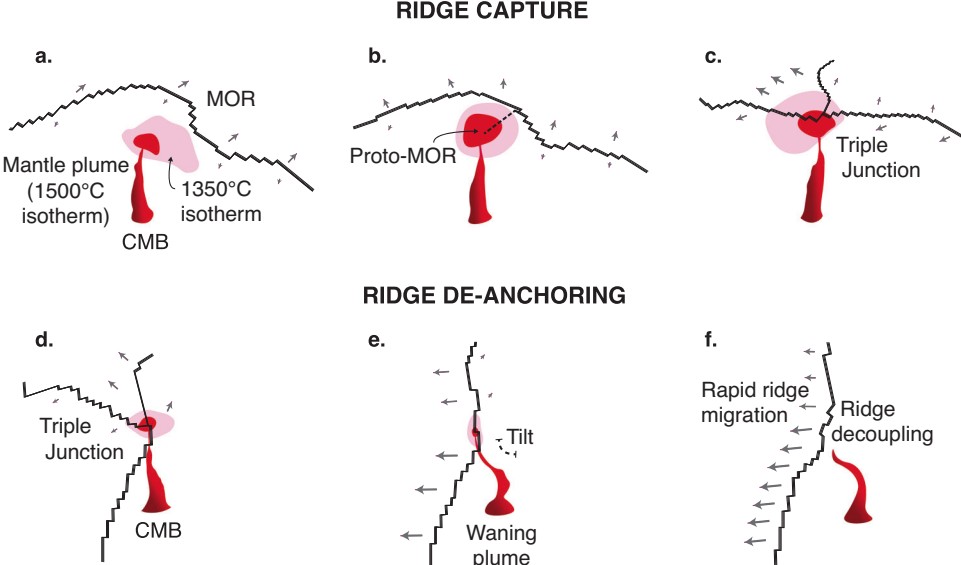

**Fig. 2 | Dynamics of ridge capture and ridge de-anchoring.** Idealised 3D illustrations of plume–ridge interactions extracted from 3D numerical model of whole-mantle convection depict three stages of ridge capture (**a**–**c**) and de-anchoring (**d**–**f**) for two separate plume–ridge interactions. **a** The closeness of the ridge–plume system is accommodated by transform boundaries progressively stepping ridge segments towards the plume and a 10° lean by the mantle plume towards the spreading ridge. **b** As the plume buoyancy flux increases (>6000 kg/s) it thermally weakens the overriding plate and develops a proto-ridge extending from the original spreading ridge; **c** spreading establishes along the proto-ridge which forms a triple junction on top of the mantle plume. **d** A triple junction is centred on a waning mantle plume (<2000 kg/s buoyancy flux) with the 1350 °C plume isotherm elongated towards the direction of relatively faster plate motion; **e** seafloor spreading is abandoned on one arm of the triple junction establishing a single plate boundary with asymmetrical spreading rates. **f** The plume wanes below 1000 kg/s buoyancy flux and decouples from the spreading ridge, which subsequently migrates away from the plume by up to 12 cm/yr. The waning mantle plume tilts 15° in the upper mantle towards the migrating ridge. CMB core-mantle boundary, MOR mid-ocean ridge. Arrows indicate the direction of plate motion.

mantle plumes that do not result in a ridge jump have low buoyancy fluxes (<2000 kg/s), a similar magnitude to the present-day Galapagos plume (1000–1800 kg/s[18,19]), suggesting that weak plumes are unable to sufficiently weaken the overriding plate to reshape spreading centres.

## Ridge de-anchoring

If high plume buoyancy flux is a significant factor controlling how spreading ridges are captured and held by plumes, then we can test the hypothesis that low buoyancy flux is associated with plume–ridge de-anchoring. From the 13 instances of plume–ridge interaction we interrogated from the numerical model, the mean plume buoyancy flux was ~25–35% lower on average during ridge de-anchoring than ridge capture (Fig. 1), indicating that reduced buoyancy flux prior to de-anchoring is consistent across all model plumes. The tight range in plume buoyancy flux across 13 instances of plume–ridge de-anchoring suggests there is a critical threshold of below ~2000 kg/s where de-anchoring is likely to occur. Only when the plume buoyancy flux wanes below this critical threshold do high convergence rates attached to one of the plates succeed in de-anchoring the spreading ridge from the mantle plume.

As before, we analysed the dynamics of all 13 plume–ridge coupled systems and present one representative example where a spreading ridge decouples from a mantle plume (Fig. 2d–f and Supplementary Movie 2). For the first 80 Myr of the time series, the plume and the ridge are coupled as the mean plume buoyancy flux gradually decreases from 2600 to 400 kg/s (Fig. 3e). This is primarily accommodated by a 50–100 °C decrease in excess temperature across the entire depth of the plume (Fig. 5e) and a reduction in the plume upwelling velocity from ~30 to 3 cm/yr in the upper mantle (Fig. 5g). As a result, thermal weakening of the overriding plate by the plume decreases, as evidenced by a 500 °C temperature drop at 30 km depth in the overriding plate (Supplementary Fig. 2d).

At 75 Myr, the plume has waned below 1000 kg/s of buoyancy flux and decouples from the spreading ridge (Fig. 3e), which rapidly migrates 600 km away from the plume over 10 Myr, with a maximum rate of 12 cm/yr (Fig. 3f). The seafloor spreading rate increases from 2 to 8 cm/yr (Fig. 3g) to accommodate rapid ridge migration following a change in plate boundary forces associated with established far-field subduction zones (Supplementary Fig. 2f). As the ridge rapidly migrates away from the plume, the conduit strongly tilts towards the ridge by up to 15° measured in the upper mantle (Fig. 3h). Transitory tilted plume conduits have been proposed to generate dual hotspot tracks[20], which could have important implications for the distribution of intraplate volcanism following plume–ridge de-anchoring.

We propose the coupling of a mantle plume with a spreading ridge acts as an anchor, thereby impeding ridge migration. De-anchoring caused by asymmetric plate velocities mechanically rips spreading ridges away from mantle plumes but is tempered by the plume buoyancy flux. The rate of ridge migration during de-anchoring is inversely proportional to the buoyancy flux (Fig. 4b), indicating that stronger plumes impede ridge migration. A stronger buoyancy flux equates to a greater upwelling velocity and higher temperature. The former means the plume penetrates to a greater extent in the overriding lithosphere, and the latter results in lower viscosity. Both are weakening agents which maintain the localisation of deformation in the overriding plate which, when reduced, facilitates ridge migration.

We contend that a reduction in plume buoyancy flux facilitates ridge migration contrary to the counter argument that the migration of a spreading ridge results in a reduction in plume buoyancy flux because (1) ridge migration occurs only after the plume buoyancy flux has substantially decreased, and (2) the plume buoyancy flux is reduced from the core-mantle boundary to the surface at the time of de-anchoring. A local decrease in buoyancy flux within the upper mantle would be anticipated following plume–ridge de-anchoring, but this would not significantly reduce the plume buoyancy flux across the entire plume conduit. We observe a wholesale reduction in the mean buoyancy flux in the leadup to ridge migration (Figs. 1 and 5h), which suggests that waning plumes facilitate plume–ridge de-anchoring.

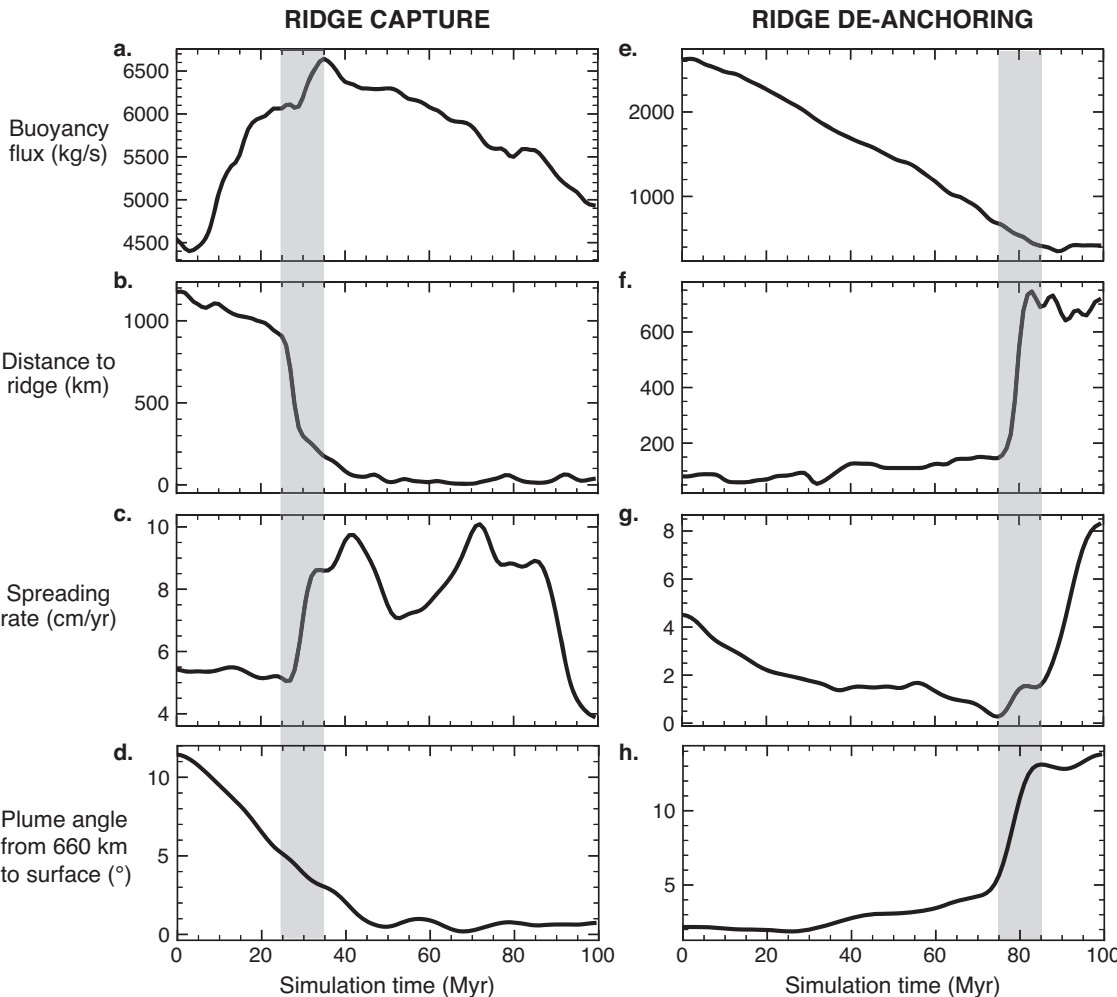

**Fig. 3 | Plume–ridge metrics for ridge capture and ridge de-anchoring.** 100 Myr time series of plume buoyancy flux, distance between the plume and ridge, and ridge spreading rates for two separate plume–ridge interactions from the 3D numerical simulation (note the *y*-axes have different scales). Shaded regions indicate the time window over which a mantle plume captures the spreading ridge (**a–d**) and where a ridge de-anchors from the mantle plume (**e–h**). **a** Buoyancy flux increases from 4500 to 6000 kg/s in the leadup to the ridge capture and peaks at 6500 kg/s as the ridge is captured. **b** The ridge and plume are initially spaced ~1000 km apart while the plume thermally weakens the overriding plate and causes the ridge to jump on top of the plume. **c** The ridge spreading rate locally increases from 5.5 to 8.5 cm/yr proceeding the ridge capture event; **d** angle of the plume conduit straightens from 11° to almost 0° at 45 Myr following ridge capture; **e** ridge de-anchoring occurs as the plume buoyancy flux diminishes from 2300 to 800 kg/s; **f** the ridge migrates away from the plume (up to 12 cm/yr) to a distance of 750 km; **g** the spreading rate of the ridge rapidly increases following ridge de-anchoring from 2 to 8 cm/yr, accelerating the rate of ridge migration; **h** the plume angle increases up to 14° towards the migrating ridge during de-anchoring.

## Reconstructing plume flux

One such example of plume–ridge coupling and de-anchoring in nature is observed between the Kerguelen plume and the SE Indian Ridge (Fig. 6). Fragmentation of the Indian, Australian, and Antarctic plates was likely driven by the initially high buoyancy flux of the Kerguelen plume which broke the Indian and Australian plates to accommodate the rapid northward trajectory of the Indian Plate propelled by double subduction[12] and/or plume push from the Reunion plume[2] (Fig. 6a). Tectonic reconstructions predict a triple junction close to the present-day location of the Kerguelen plume[5,21], and the eruption of multiple LIPs associated with the Kerguelen plume[5] suggests a buoyancy flux at least as high as the triple junction-forming plumes observed in our numerical model (Fig. 2c).

The coupling of the Kerguelen plume with the SE Indian Ridge produced the Ninety-East Ridge on the Indian Plate from >80 to 43 Ma (Fig. 6b). The Ninety-East Ridge records ~37 Myr of plume products embedded within the Indian plate from the coupled Kerguelen plume, up until plume–ridge de-anchoring at 43 Ma. We relate variations in plume flux from the Kerguelen plume to magma production along the Ninety-East Ridge by measuring the fluctuations in seafloor bathymetry at multiple cross-sections (Fig. 7a–d). This involves subtracting the height of the Ninety-East Ridge from the height of adjacent seafloor away from plume–ridge influence and integrating the area (Supplementary Fig. 3, see "Methods" section). The resulting magma production (in km²) is the excess area (over regional bathymetry) that includes all plume material incorporated into the Indian Plate from the Kerguelen plume, such as erupted volcanics, sill injection, and magma underplating. This excludes any plume products on the slower-moving Antarctic or Australian plates.

From our calculations, the magma production increased from 220 km² at 6° N (78 Ma) and reached its peak of ~440 km² at 15° S (55 Ma) implying the buoyancy flux for the Kerguelen plume reached its zenith during its coupling with the SE Indian Ridge (Fig. 7b–d). Since then, the magma production has decreased to a minimum point of 160 km² at 27° S (45 Ma) after which the SE Indian Ridge decoupled from the Kerguelen plume (43 Ma). Superimposing box plots of radiogenic isotopes on top of Fig. 7b–d show a strong association with the estimated magma production for a given latitude. In general, the

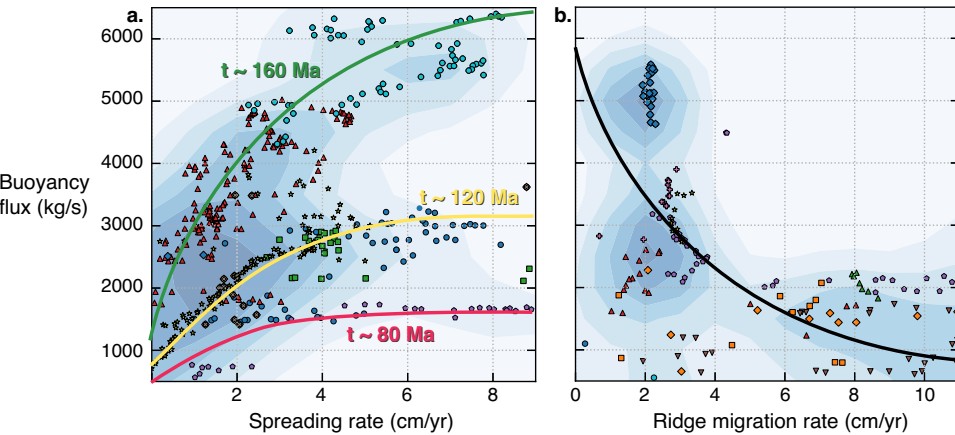

**Fig. 4 | Relationship between plume buoyancy flux, local spreading rate, and migration rates.** Summary relationships for 13 plume–ridge interactions during their coupling (**a**) and de-anchoring (**b**). Each plume–ridge coupled system is annotated by a unique marker and colour. The background colour is a kernel density estimate based on the density of points. **a** Buoyancy flux and the local spreading rate along ridges are positively correlated for coupled plume–ridge systems in the numerical model of whole-mantle convection. The approximate longevity of 13 coupled plume–ridge systems is annotated on the axes. Plumes that are active over a longer timespan within the numerical model tend to have a larger buoyancy flux, and vice-versa, suggesting that strong plumes are generally established over longer timespans than weak plumes. The longevity of plumes modulates the correlation between buoyancy flux and local spreading rate. **b** Buoyancy flux and rates of ridge migration within a 20 Myr time window of de-anchoring are inversely correlated. The curve shows how ridges move faster away from mantle plumes with lower buoyancy flux.

variations in magma production with Sr, Nd, and Pb isotopic compositions corroborate a trend from moderately low plume flux at 78 Ma to high plume flux at 55 Ma and low plume flux at 45 Ma. This implies that the plume buoyancy flux diminished significantly prior to the de-anchoring of the Kerguelen plume from the SE Indian Ridge at 43 Ma.

Isotope ratios help to discern the mantle source of melting for eruption products. Geochemical variations in mantle domains between mid-ocean ridge basalts (MORB) and ocean island basalts (OIB) have long been attributed to heterogeneity and mixing between upper and lower mantle reservoirs[22]. For a trail of volcanic products linked to the same plume–ridge system, the compositional spread between MORB and enriched plume end members likely reflects changes in the melt fraction delivered by the plume from more deeply sourced upwelled mantle[23]. It follows then that the plume buoyancy flux should play a first-order control on the mixing between these two end members[24]. Additional variations in enrichment may occur due to the entrainment of upper mantle into the upwelling plume or variations in plume source composition with time. Using the end-member compositions of enriched plume compositions defined by 24–25 Ma basalts and picrites of the Kerguelen Plateau and normal mid-ocean ridge basalts (N-MORB) from the SE Indian Ridge[25], we compare the relative proportions of plume and asthenospheric mantle along the Ninety-East Ridge to changes in plume buoyancy flux (Fig. 8).

Lower $^{87}Sr/^{86}Sr$ and higher $^{143}Nd/^{144}Nd$ characterise the youngest samples along the Ninety-East Ridge (Fig. 8a, d) reflecting greater proportions of upper mantle melting. Pb isotopes exhibit a weaker age correlation (Fig. 8c). In general, the variations in Sr, Nd, and Pb isotopic enrichment illustrate a trend from higher plume proportions at 55 Ma to higher N-MORB proportions at 45 Ma, reflecting a diminished buoyancy flux of the Kerguelen plume immediately prior to plume–ridge de-anchoring at 43 Ma. This coincides with the onset of rapid northward migration of the SE Indian Ridge from the Kerguelen plume driven by the fast motion of the Australian Plate (Fig. 7e).

### De-anchoring, a driver of ridge migration?

The decrease in magma production and higher upper mantle proportions in melts at the youngest part of the Ninety-East Ridge support our hypothesis that the waning Kerguelen plume culminated in the de-anchoring of the SE Indian Ridge at 43 Ma. We propose that thermal weakening of the overriding plate likely decreased as the plume buoyancy flux waned, thus the localisation of deformation could not be sustained while strong asymmetric plate velocities associated with the rapid migration of the Australian plate mechanically ripped the SE Indian Ridge away from the Kerguelen plume. As a result, the SE Indian Ridge has migrated away from the present-day location of the Kerguelen plume at 2.7 cm/yr (Fig. 7e), translating to 5.4 cm/yr northward migration of the Australian plate[26], and a transition to intraplate volcanism on the Kerguelen Plateau[14]. We speculate that if the buoyancy flux of the Kerguelen plume were greater at 43 Ma, then progressive ridge jumps closer to the Kerguelen plume would be observed.

Following de-anchoring, the volcanic products should switch from MORB to OIB systematics as intraplate volcanism is established. Since its separation from the SE Indian Ridge, the Kerguelen plume has produced the intraplate Kerguelen Archipelago on the northern Kerguelen Plateau with a range of flood basalts aged between 29 and 24 Ma[14]. Low eruption volumes for these islands indicate a waning plume buoyancy flux (827 kg/s) with volcanism occurring over a diffuse area[14]. Present-day estimates of buoyancy flux for the Kerguelen plume is between 200 and 500 kg/s[18], which is well below the 1000 kg/s threshold for our predictions of plume–ridge de-anchoring to take place and suggests that the Kerguelen plume has continually waned from 60 Ma to the present day. Our numerical model predicts that following de-anchoring, the waning plume can tilt 15° within the upper mantle towards the migrating ridge. If the stem of the Kerguelen plume is currently beneath Heard/McDonald Island, then both the Kerguelen Archipelago and Amsterdam–St Paul Islands could be diapirs rising from the plume conduit tilting northwards towards the SE Indian Ridge. This would explain why the isotopic compositions of Amsterdam–St. Paul Islands are strongly N-MORB (from the SE Indian Ridge) mixed with a small enriched plume component[27].

If the buoyancy flux of mantle plumes is a significant factor controlling ridge capture and de-anchoring, then this mechanism may be used to help explain the evolution of plume–ridge interactions and asymmetric accretion of oceanic crust globally. We find the attraction between mantle plumes and spreading ridges is strongest for plumes of high buoyancy flux. For example, up to 300 km displacement away from the expected age-progressive seamount chains has been attributed to the attraction of the Hawaiian plume (present-day buoyancy flux of 7400 kg/s[18]) towards the ridge separating Kula and Pacific

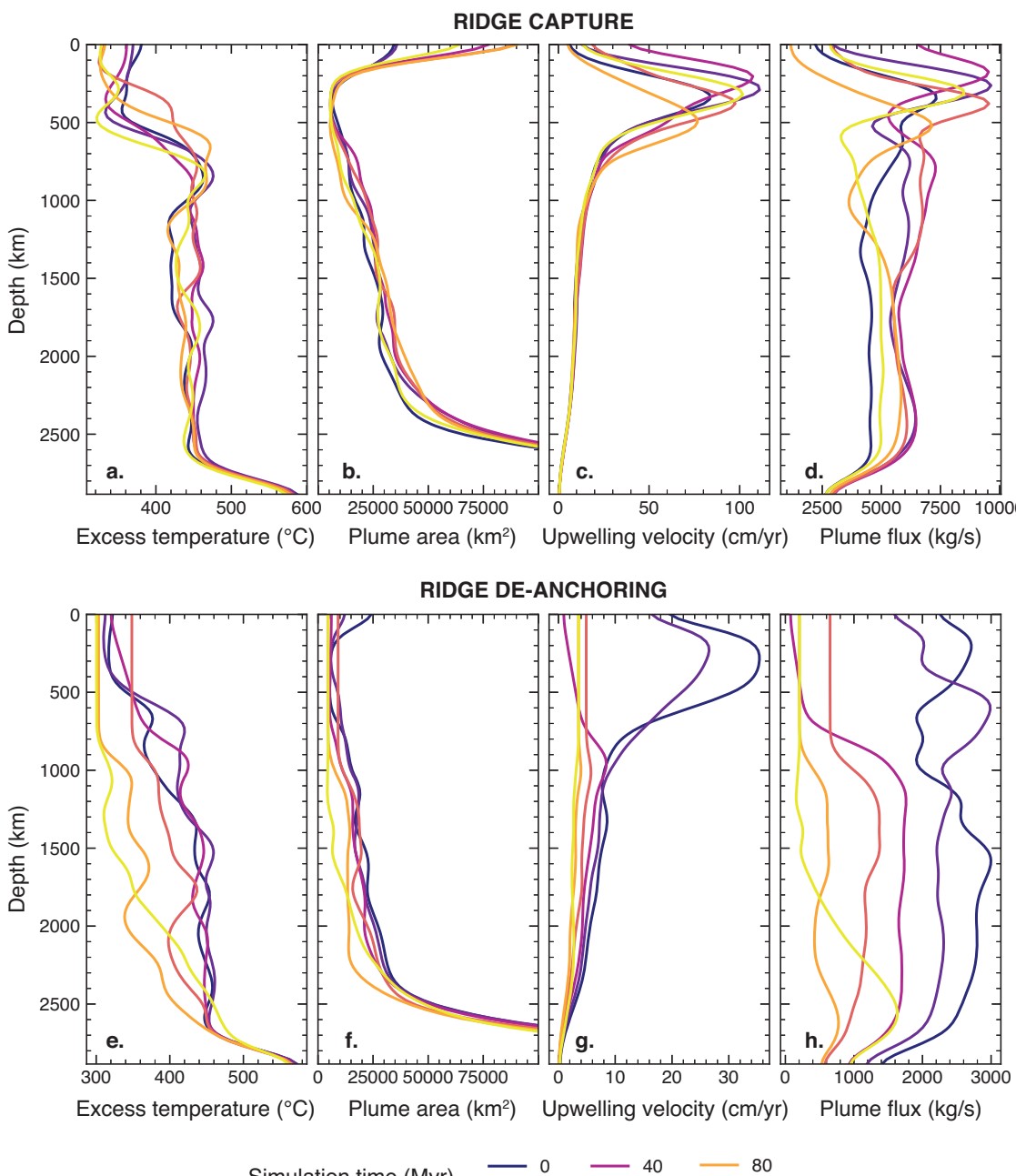

**Fig. 5 | Plume attributes with depth during ridge capture and de-anchoring.** The key variables used to calculate the plume buoyancy flux (excess temperature, plume area, upwelling velocity) are plotted with depth for ridge capture (**a**–**d**) and de-anchoring (**e**–**h**) in two separate plume–ridge interactions (note the *x*-axes have different scales). These correspond to the plume–ridge interactions documented in Figs. 2 and 3. The evolution of each variable is captured by a unique line colour representing the simulation time from 0 to 100 Myr in 20 Myr increments.

plates[28]. High plume buoyancy flux may also be a key driver of triple junction formation, since a faster upwelling velocity and excess temperature will fragment the overriding plate into smaller plates to accommodate more efficient plate motion.

For waning mantle plumes close to the critical 1000 kg/s buoyancy flux threshold, spreading ridges may abandon the plume. The balance of forces on each plate pair (or triple) controls the rate of ridge migration. Strong asymmetries in plate boundary forces are a prerequisite to drive rapid ridge migration enabled by plume–ridge de-anchoring. For the Kerguelen plume, strong slab pull on the Australian plate married with little to no plate boundary forces on the Antarctic plate translated to the rapid northward migration of the SE Indian Ridge during de-anchoring.

If the balance of forces on each plate pair is similar to each other, then the spreading ridge would migrate much more slowly following plume–ridge de-anchoring than if the plate force balance was highly asymmetric. The Atlantic Ocean is not bounded by subduction zones which indicates the balance of plate boundary forces along the Atlantic Ridge is relatively similar to each other[29]. Therefore, it is unlikely that plume–ridge de-anchoring would translate to fast ridge migration rates. For example, the transition from the Walvis Ridge to the intraplate Tristan-Gough seamount chains reflects the de-anchoring of the mid-Atlantic spreading ridge from the Tristan plume due to decreased plume buoyancy flux at around 70 Ma[30] (Supplementary Fig. 4), which did not coincide with rapid ridge migration. Instead, de-anchoring is merely manifested by a gradual separation between plume and ridge.

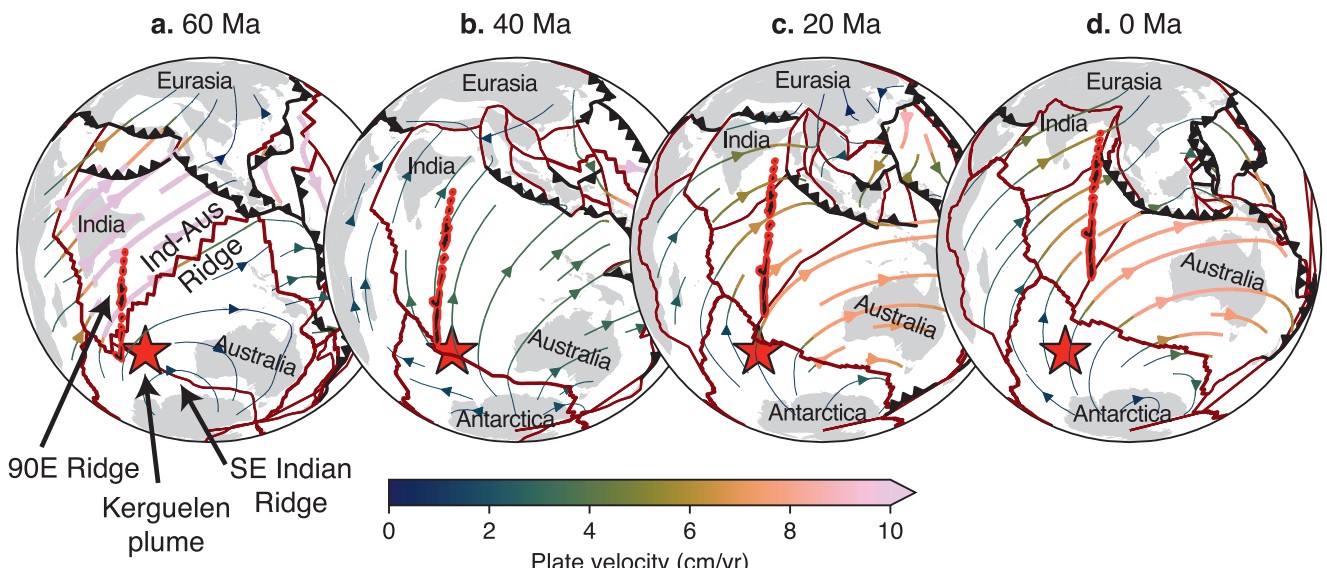

**a. 60 Ma**   **b. 40 Ma**   **c. 20 Ma**   **d. 0 Ma**

Plate velocity (cm/yr)

**Fig. 6 | Plate reconstruction of de-anchoring between the Kerguelen plume and SE Indian Ridge.** The location of the Kerguelen plume is fixed to its present-day coordinates of 69° W, 50° S; plate boundaries and velocities were reconstructed at 20 Myr increments from 60 Ma to the present day[26] using GPlately[40]. **a** The Kerguelen plume was centred on a triple junction separating Indian, Australian, and Antarctic plates as India is propelled towards Eurasia by double subduction[12]. **b** Following plume–ridge de-anchoring, the SE Indian Ridge began to migrate northward at 2.7 cm/yr and spreading ceased along the India–Australia ridge. **c** Increased spreading rates along the SE Indian Ridge at 20 Ma translated to northward motion of the Australian plate at 7 cm/yr. **d** Present-day configuration of plate boundaries and the location of the Kerguelen plume.

Similarly, the force balance between the Nazca and western Pacific plates is mostly comparable (convergence rates between 9 and 11 cm/yr, along the Andes and Tonga-Kermadec Trench[26]), which explains why the East Pacific Rise has not migrated rapidly away from the Easter plume following plume–ridge de-anchoring[31]. The seamounts of the Easter hotspot chain have become progressively less enriched westwards towards the present-day location of the Easter plume[32], suggesting its buoyancy flux has continually waned below that of the Galapagos plume[33].

Our work demonstrates that the interactions between plumes and ridges are modulated by plume buoyancy flux which can anchor plate boundaries in place. This outcome provides new insights into the forces that govern plate tectonics and plume dynamics, which has significant implications for the evolution of plate boundaries, the distribution of plume-related volcanism, and the reconstruction of ocean basins.

## Methods

### Setup of a whole-mantle convection model

We used the StagYY code[34] to model incompressible whole-mantle convection using the Boussinesq approximation for an incompressible fluid in 3D spherical geometry. We set the reference Rayleigh number to $10^7$, which favours the self-generation of Earth-like amplitudes of surface topography (-20 km peak-to-peak), heat flow (-50 TW), and plate velocities (-3 cm/yr). Viscosity varies with both temperature and depth through an Arrhenius law, producing 7 orders of viscosity variations throughout the mantle[16]. To model lower convective vigour in the lowermost mantle, thermal expansivity decreases by a factor of 3 throughout the mantle domain, consistent with mineral physics experiments at high pressure and temperature[35], and we impose a 30-fold increase of viscosity at the transition zone, as suggested by the study of Earth's geoid[36]. Self-consistent plates are generated at the surface through pseudo-plasticity, by using a surface yield stress of -50 MPa and a depth-dependent yield stress coefficient of -1000 Pa/m. Once the yield stress is reached, viscosity drops by several orders of magnitude, focussing deformation into narrow regions and thus mimicking plate-boundaries formation, distribution, and evolution

self-consistently with mantle convection[37]. We model continents as less dense, more viscous, and stronger rafts than oceanic lithosphere in order to ensure their positive buoyancy and stability throughout the model evolution. Their initial geometry and location correspond to a reconstruction of Earth's continents at 80 million years ago[38], reflecting their dispersed geographical distribution at that time. Continental material is tracked by tracers, which are allowed to advect after an equilibration phase (not considered for the here-described model analysis) during which convection reaches a quasi-statistic equilibrium, corresponding to statistically stable average mantle temperature and boundary heat fluxes.

The model used here corresponds to Model 2 of ref. 16, where all quantitative details can be found about its setup, the general behaviour of mantle plumes, and the method used to track absolute plume motions.

### Calculation of plume buoyancy flux

The buoyancy flux of mantle plumes is calculated from the following relation[17],

$$B_p = \rho_m \alpha \Delta T A_p v_p \qquad (1)$$

where $\rho_m$ is the mantle reference density, $\alpha$ is the reference thermal expansivity, $\Delta T$ is the mantle plume excess temperature, $A_p$ is the cross-sectional area of the mantle plume conduit, and $v_p$ is the buoyant rising speed. Plumes are identified from the numerical model by isolating where the temperature is greater than one standard deviation from the mean temperature for each depth interval from the core-mantle boundary to the surface of the model. The excess temperature is then calculated across this temperature where the plume can be resolved (e.g. some plumes terminate below the surface of the Earth). The cross-sectional area is calculated from the area of the convex hull of points that encircles the plume for each depth interval, and the upwelling velocity is extracted from the radial component of the velocity field in spherical coordinates within this convex hull. The plume buoyancy flux is calculated using the above relation and averaged

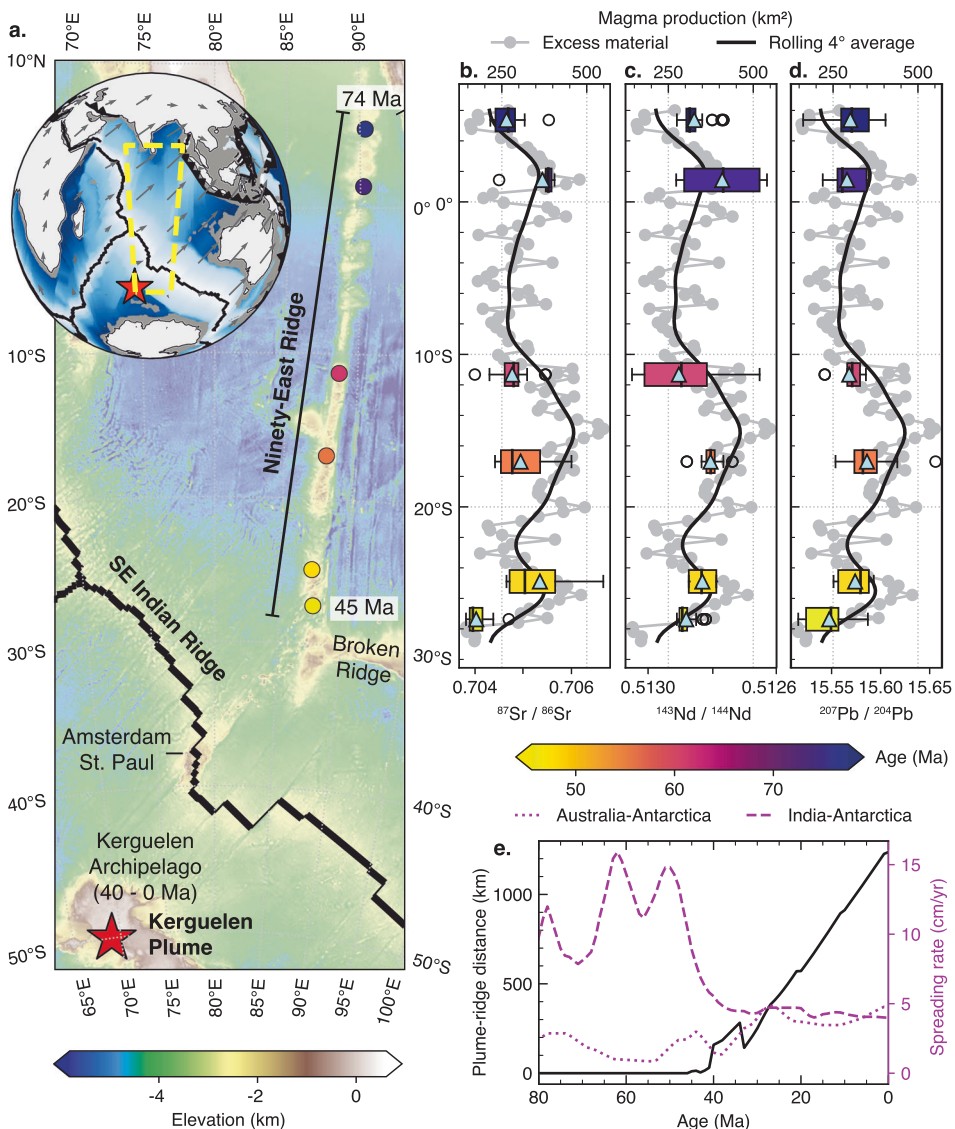

**Fig. 7 | Magma flux and radiogenic enrichment along the Ninety-East Ridge.** The Kerguelen plume was coupled to the SE Indian Ridge from >78 Ma to 43 Ma, producing the extensive volcanic products that comprise the Ninety-East Ridge. **a** Map of seafloor elevation from ETOPO1[41] showing the position of the Kerguelen plume, the geometry of the SE Indian Ridge, and age progression of volcanic products which comprise the Ninety-East Ridge. Magma production calculated at multiple latitudinal cross-sections along the Ninety-East Ridge (see "Methods" section), overlain with isotope ratios of **b** $^{87}Sr/^{86}Sr$, **c** $^{143}Nd/^{144}Nd$, and **d** $^{207}Pb/^{204}Pb$. Box plots are coloured by age, black lines indicate the median, blue triangles indicate the mean, and white circles indicate outliers. **e** Distance between the Kerguelen plume and SE Indian Ridge since 80 Ma, and rate of seafloor spreading between Australia, India, and Antarctica calculated from a plate reconstruction[26] using a fixed plume reference frame.

across the depth range of the plume. Plume tilting is calculated from the difference between the location of the centroid of the cross-sectional area at each depth interval and the mean plume centroid averaged across the entire depth range of the plume. In the main text, the tilting is calculated in the upper mantle (from 660 km depth to the surface, where tilting is generally greatest) and expressed in degrees by summing the distance away from the mean plume centroid and calculating its arctangent with 660 km to the surface. The plume buoyancy flux within the numerical simulations of whole-mantle convection ranges from 10 to 20,000 kg/s and is 5000 kg/s on average[16]. A plume tracking algorithm was implemented to track the migration of plumes (relative to the geographic grid) and the changes in buoyancy flux across multiple timesteps. If a plume is spaced <400 km between timesteps (measured at the surface), they are considered to be the same plume. Plumes move little compared to tectonic plates between model timesteps, which means the distance tolerance of 400 km relative to the geographic

grid is sufficiently large to identify separate plumes yet small enough to avoid duplicate plumes. If multiple plumes are detected within 400 km radius of each other (and have different plume fluxes) then these are tracked as separate plumes.

**Calculation of ridge spreading rates and distances to plumes**
Spreading ridges are identified in the numerical model from the surface temperature and velocity fields. We identify sharp discontinuities in the surface velocity field combined with a white top-hat transform of the surface temperature (a peak-finding algorithm) to identify continuous ridge segments for each timestep. (Using just the velocity incorrectly identifies triple junctions, and using only the temperature incorrectly identifies plumes as spreading ridges.) We calculate the spreading rate by seeding two points orthogonal to the strike of the ridge and sampling the velocity field at those locations. The distance to the nearest plume is calculated using the Haversine distance, which accounts for the curvature of the sphere.

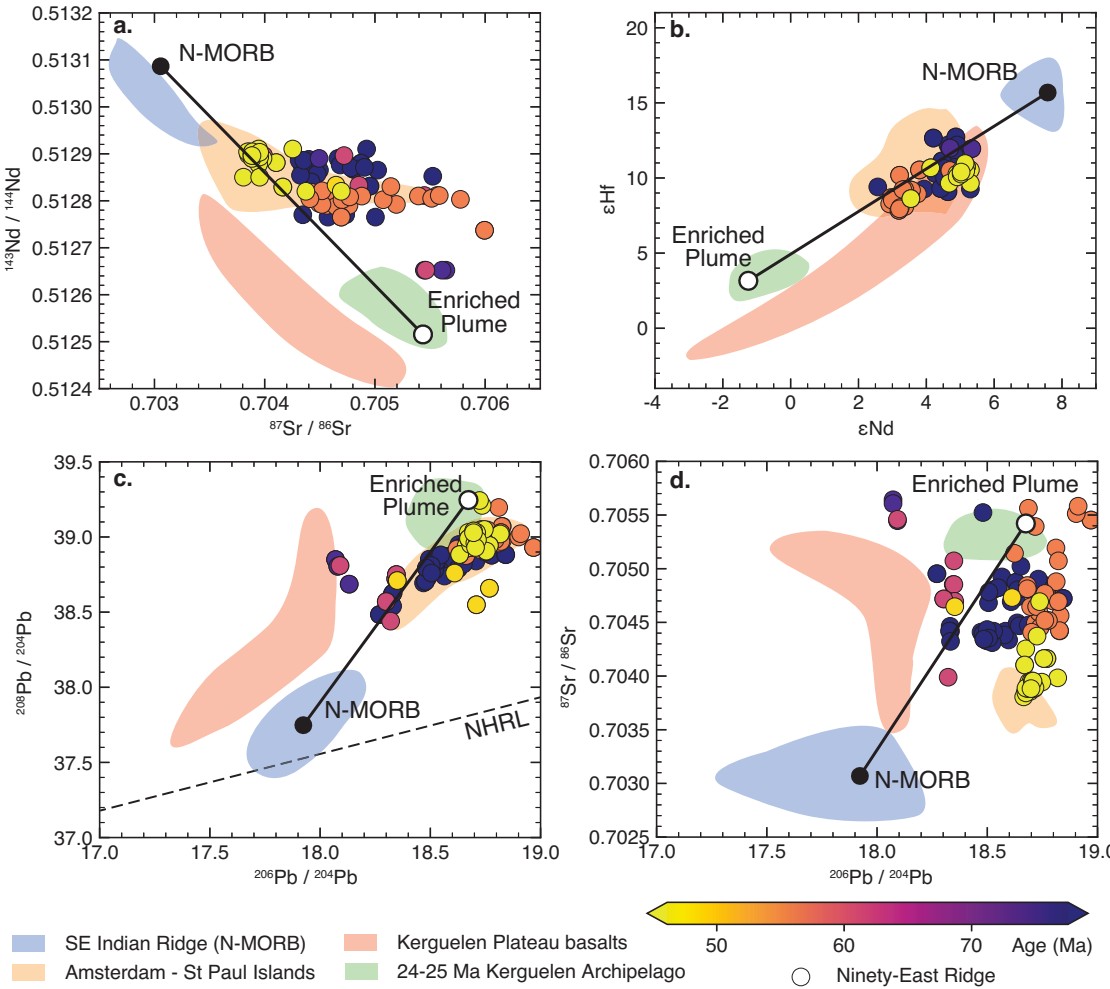

**Fig. 8 | Sr–Nd–Hf–Pb isotope variations for basalts along the Ninety-East Ridge.** Samples along the Ninety-East Ridge were compiled from various sources[42,43] and are coloured by their eruption age. Comparisons to other plume–ridge systems and ocean island basalts are indicated by the coloured regions. The end-member compositions of enriched plume and normal mid-ocean ridge basalt (N-MORB) are from 24 to 25 Ma basalts and picrites from the Kerguelen Plateau and the SE Indian Ridge, respectively[25]. Amsterdam–St Paul Islands and Kerguelen Plateau basalts are plotted for regional comparison. **a** $^{143}Nd/^{144}Nd$ plotted against $^{87}Sr/^{86}Sr$ shows a lower concentration of Sr for youngest samples along the Ninety-East Ridge aged ~45 Ma trending towards N-MORB. **b** εNd against εHf shows a spread between N-MORB and enriched plume across multiple samples. **c** $^{208}Pb/^{204}Pb$ plotted against $^{206}Pb/^{204}Pb$ shows more radiogenic Pb for younger sections of the Ninety-East Ridge (NHRL Northern Hemisphere Reference Line). **d** $^{87}Sr/^{86}Sr$ plotted against $^{208}Pb/^{204}Pb$ exhibit less Sr enrichment for a relatively uniform Pb trending towards N-MORB.

## Calculation of convergence rates and distances to plumes

Subduction zones are identified in the numerical model from the density field at 125 km depth and the surface velocity field. We identify sharp discontinuities in the surface velocity field combined with a white top-hat transform of the density field to identify continuous trench segments for each timestep. We calculate the convergence rate by seeding two points orthogonal to the strike of the trench and sampling the velocity field at those locations. The distance to the nearest plume is calculated using the Haversine distance, which accounts for the curvature of the sphere.

## Estimates of magma flux for the Kerguelen plume

Latitudinal cross sections of ETOPO1 elevation were taken at 0.3° increments along the Ninety-East Ridge from 6° N to 27° S. Each cross section is 3° in length along the longitudinal plane. Sediment thickness is subtracted from the topography to remove the influence of continental sediments on the calculation of magma flux[39]. The area of the topographic bulge is calculated by selecting a basal seafloor height for each cross section, which we define as the 5th percentile of the elevation across the cross section. The area is calculated by integrating the elevation above the seafloor basal height (Supplementary Fig. 3).

Choosing a different seafloor basal height changes the magnitude of the area calculation but does not perceptibly alter the relative differences in area across multiple cross-sections. We interpret these variations in area between cross sections as caused by changes in plume magmatic flux. This encompasses all excess material emplaced in oceanic seafloor from the mantle plume, including erupted volcanics, sill injection, and magma underplating. Here we assume eruption material was incorporated instantaneously in oceanic seafloor (i.e. no overlap of eruption ages between cross sections), and the age of eruption material is identical to the age of the seafloor (which is reasonable during periods of plume–ridge coupling because the formation of the Ninety-East ridge occurs at the spreading ridge). More advanced methods to calculate the absolute magma flux for a given cross section would account for isostatic loading of the lithosphere to estimate the magma flux below the surface of the seafloor, however, this is not critical for our study because we are interested in long-wavelength trends and not the absolute values of magma flux.

## Data availability

The findings presented in this paper are based on data sourced from previously published works, as cited in the text. The numerical model

of mantle convection was originally published in Arnould et al.[16] and the geochemistry data were sourced from Mahoney et al.[42] and Frey et al.[43].

## Code availability

Plate reconstruction software, GPlates and GPlately, is freely available from www.gplates.org/download.html and https://github.com/GPlates/gplately, respectively. Thermodynamic modelling software, Perple_X, and associated thermodynamic databases are freely available from www.perplex.ethz.ch. The mantle convection code StagYY is the property of Prof. Paul Tackley and Eidgenössische Technische Hochschule (ETH) Zürich. It is available on request by P. Tackley (paul.tackley@erdw.ethz.ch). Minor edits of the code can be provided upon request to M.A. and N.C.

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

## Acknowledgements

This study was supported by the AuScope Simulation, Analysis & Modelling node funded by the Australian Government through the National Collaborative Research Infrastructure Strategy, NCRIS. This research benefited from the assistance of resources from the National Computational Infrastructure (NCI), which is supported by the Australian Government. We gratefully acknowledge funding from the Australian Research Council through grants DP200100966 (M.S. and S.W.), and DP180102280 (J.W. and S.W.); support from the Australian Antarctic Science Program Grant 4598 to J.W., M.S., and R.C.; M.A. benefited from funding from CNRS INSU PNP 2021 and from Univ. Lyon 1 BQR 2021.

## Author contributions

B.M. compiled and interpreted the trace element and radiogenic isotope geochemistry, interrogated the model of whole-mantle convection and coupled the thermodynamic modelling framework, wrote the paper, and prepared the figures. M.S., S.W., J.W., and R.C. conceived and directed the project. M.A. and N.C. ran the model of whole-mantle convection and analysed their output. R.D. contributed to the analysis of geochemical trends in plume–ridge systems. All authors participated in the discussion and interpretation of the results and in the preparation of the manuscript.

## Competing interests

The authors declare no competing interests.
