## [Peer Review File · Nature Communications]

REVIEWER COMMENTS

Reviewer #1 (Remarks to the Author):

This manuscript addresses a topic that has been in need of attention for quite some time. Considerable research has considered plume-ridge interaction from geochemical, geophysical, and geodynamics perspectives, but those studies have not directly addressed the specific mechanism of how plumes interact with ridges and affect their migration via anchoring/de-anchoring. The multidisciplinary approach taken by the authors that combines geodynamic modeling with geochemistry to “ground truth” the modeling is an effective one, for the most part, but not as clearly conveyed or applied as it could be in the manuscript, something easily addressed. The idea that plume flux into a ridge system is a controlling factor in terms of how effectively a ridge can be anchored to a plume is an important concept to have identified in a systematic way. In short, I anticipate that this paper will be appreciated and referenced by many researchers working on near-ridge plumes and near-plume ridges in the future.

Some comments as I go through the manuscript:

1. I am not a modeler, so I cannot assess the model details or methodology. That said, it would help readers like me who are not modelers to clarify a bit of the Results section. For instance, why was a 323 Myr timespan chosen? At the most basic level, is this modeling a generic spherical planet with Earth-like mantle convection or one that has some of Earth's characteristics (plates, plumes, etc.)? Stating this unequivocally up front would eliminate some questions to those who are not in the modeling realm, and would help make this more friendly to the non-modeler.
2. Figure 1 could use a bit more explanation in the caption of how the flux was modeled; just recapping it in a sentence in the caption would help make this a more understandable figure.
3. Figure 2 would benefit from some explanation in the caption to indicate that it's effectively showing 3 dimensions; at first it is confusing b/c it looks like it's a bird's eye (2D) view, but I think (?) that it's meant to show the tilt in the plume conduit? If this is incorrect, then it supports my point that it's a bit hard to know how to understand this figure clearly. Same applies to the two animation clips.
4. In Results (and in the model), is the plume flux changing in response to the interactions with the ridge, or is it changing b/c it's a generic plume lifetime that includes increasing in intensity then waning flux as it goes extinct? What I'm trying to ask is whether plume flux is tied to ridge motion in some way, because around line 94, the paper reads as though the flux is being documented, not pre-determined. I am sure I've missed something here, but stating clearly (even if it's obvious to modelers) that you are or are not modeling the plume flux behavior would help. This section is written as though the plume flux is a response to the model parameters, but I'm unclear if that's the case in reality. Again, if I missed something, apologies. NOTE: I think I sorted this out, see point 21...if that's correct, some of these comments may not be as relevant; if that's NOT correct, then definitely things need to be clarified.
5. Line 96: Is this meant to be Figure 2a or 3a? Seems confusing to be jumping to Figure 3 when the paper is mostly focused on Figure 2 here.
6. Why does the spreading rate increase again (Figure 3g) once the plume has been disconnected from the ridge? This seems counter-intuitive from a physical perspective and definitely needs to be discussed.
7. The Results section would be easier to follow if it were split into shorter, single-focus paragraphs, each dealing with one figure/phenomenon at a time.
8. Line 113-114 how old is “older” seafloor? How much older can it be in a near-ridge plume system?

9. Line 118: Within 1 Myr of what?
10. Line 121: Need a “within” in front of 1000 km, and the “which” should be a “that”
11. Caption of Figure 4: Isn't it kind of a given that “plumes that are active over a longer timespan within the numerical model tend to have a larger buoyancy flux” or is that not the case? Would be good to clarify this statement further. Furthermore, is this a finding, or a constraint of the model?
12. Paragraph beginning line 146: This pgh can use reworking to clarify its content. The part beginning “from the 13 instances of...” conveys an important finding, but it is delivered after explaining why that might be the case, so it's hard to follow. Also, are plumes being considered that do not get to the surface? That's implied in line 151 with the plume reaching a “shallower depth.”
 - a. And how does plume viscosity matter here; if the ridge is being anchored, it's presumably somehow related to the mantle's viscosity/physical properties; what is the role of the plume's viscosity in this system?
13. Figure 6: Should label the India-Australia ridge.
14. Line 176: What is the justification for the “most extensive plume products” mentioned here? Volume? This needs a citation at the minimum and ideally some graphic/explanation of how this is known, given how impactful this evidence could be for the argument laid out in this paper.
 - a. And why would a tilted plume result in more extensive plume products, wouldn't more extensive material result from higher fluxes, not plume angle? Or is more widespread plume materials the appropriate description (as it's described in the final paragraphs)?
15. Line 185: I'm not so sure that the ratio between MORB and more enriched plume compositions reflects changes in the capacity of the plume to delivery more deeply sourced melt to the surface so much as on whether the plume is diluted by entrained upper mantle and/or depleted lower mantle material. The Zindler et al. (1979) reference predates a lot of research into this topic that has happened since; this warrants clarification/revisiting/updating.
 - a. Similarly, in the next sentence, stating that plume buoyancy flux plays a first-order control on the spread between depleted and enriched endmembers is also at the minimum an oversimplification, and really more of an inaccurate generalization.
16. Line 190: the first $^{143}\text{Nd}/^{144}\text{Nd}$ should presumably be $^{87}\text{Sr}/^{86}\text{Sr}$?
17. It would really clarify this isotope argument to produce a simple timeline of the different isotopic ratios in samples, instead of the x-y isotope plots. It's very hard to piece this together to understand the proposed pattern.
 - a. Also, why include N Atlantic Ridge (is that meant to be Australian?), or Pitcairn if they are ancillary to the story? Simpler would be better.
18. The model could use clarification and simplification in how it's explained. Is the idea that when the plume is anchoring the ridge, we should see more intermediate signatures along the two-component spectrum, but more plume-like signatures when the plume is on its own/ridge-free? (which is pretty much the basis of most plume-ridge interaction models that use geochemistry as the metrics)
 - a. If so, the PCA doesn't add much to this; seems like you don't need it here to test your hypothesis. PCA kind of muddies the water since the only focus is on PC1, which is just two-endmember mixing. What is involved in PC2, then, and why is it needed at all? Why not just use average plume and average MOR compositions (as is done in the binary isotope plots already)?
19. Again, similar to point 18 as I try to work this out: If I am understanding the proposed isotopic mixing model, we should be seeing more ridge-like compositions first along the 90E ridge's oldest samples, then a shift to more plume-like/intermediate signatures, then back again to ridge only for the youngest

ones...right? This model could be conveyed much more clearly if that's what it is; if not, then it definitely needs more explanation; the reliance on PCA to get the point across isn't that effective. It would also be much more clearly conveyed in a simple graphic with time along one axis, as stated above, than forcing the reader to decipher it here by color (Figure 7).

a. Also, the model works ok for (a) and (b) in Figure 7 (Sr, Nd, Hf isotopic ratios), but it really doesn't work well for either $^{206}\text{Pb}/^{204}\text{Pb}$ - $^{208}\text{Pb}/^{204}\text{Pb}$ (c) (why don't the younger samples get more MORB-like than they do, but stay closer to the plume composition?), and why do the older samples in the Sr-Pb system (d) not sit closer to N-MORB compositions? The model should work in all isotopic systems, but it doesn't from what I can see, which calls it into question and warrants attention.

b. Similarly, if this is an issue of going from ridge-only to plume/ridge mixing to ridge-only again, why are there so many inflection points in magma flux in Figure 9(b)? What happened to cause those complexities in the magma flux vs latitude (time) curve? Same question applies for the Walvis/Tristan analysis in the supplemental docs.

20. Line 203: What's "magma flux" referencing here? In the plume or the ridge, and how does it differ from buoyancy flux? I assume it's MOR flux?

21. Ok so maybe the issue I'm having is that the authors mean buoyancy flux/plume contribution to the ridge whenever it is mentioned, not in fact changing plume flux over time? If so, it took me until nearly the end of the paper to understand that nuance, despite careful reading. Some simple clarification early on to indicate that the study focuses on the RIDGE's erupted products and the buoyancy flux of magma to the ridge (from both the conventional MOR upwelling plus or minus plume contributions) would probably sort out a lot of the potential for confusion. I'm sure it's mentioned early on, but as someone who works on plumes, I definitely assumed that "buoyancy flux" referenced the PLUME's buoyancy flux, not the flux of material to the MOR throughout this paper. My error if I missed it early on, but many people read papers far more quickly, so finding a way to make it bomb-proof what you are modeling early on in the paper would be great (just bear in mind that the use of "buoyancy flux" is most commonly and automatically attributed to the PLUME among plume researchers, not the supply to the ridge).

22. Line 246: Really can't follow what's meant here as written. Could use clarification.

23. Are there any implications of this model for the ocean islands and their compositions (and/or morphologies/sizes) being formed by the plume in a near-ridge environment?

24. In general, the manuscript would benefit from a detailed proofread that focuses on clarifying sentences, finding a more systematic way to convey information via figures (it bounces around between figures quite a bit, which makes it frustrating to read), designing shorter paragraphs focused on single topics, and use of commas and other minor grammatical tools to help clarify sentences (e.g., line 146 needs a comma after "anchor" to make it read easily).

Reviewer #2 (Remarks to the Author):

I have reviewed the manuscript 'Spreading ridge migration enabled by plume-ridge de-anchoring', by B. Mather and coauthors. This manuscript uses numerical models of viscous flow to draw conclusions about the dynamics of ridge migrations and the controls operated by ridge-plume interactions.

The topic of the study is one of interest and has the potential to advance tectonic reconstructions at the global scale. However, I would see this study more suited for a specialized outlet. In fact, the study revolves around analyses of model behavior upon changes of the input parameters. I fail to see a testable hypothesis that is rooted in specific observations, and that is tested against independent data and uncertainties. The study makes a comparison with SE Indian Ridge reconstructions, but to me this seems more a case of resemblance rather than prediction. This is not to say there is no merit in this study. Rather, that it does not make a convincing case for a paradigm shift in geological knowledge.

Below are a few points that authors may find useful:

That plate tectonics is primary driven by slab pull is a rather partial view: there is a significant body of work illustrating the relevance of controls on plate tectonics operated by pressure-driven mantle convection (for instance, the work of Lenardic, Richards and coauthors).

Some of the statements that are key to the dynamics proposed in this study are based on other studies (ref. 2 for plume push, ref. 11 for slab pull) that contain proposals rather than tested hypotheses. This gives the impression that these are well established dynamics, which is not the case.

A notion invoked at line 105 is that a strong plume can break the lithosphere. This statement should be backed up with either references (the past work of Roger Buck may be something to look at) or tests/calculations. For instance, can the push of a plume exceed the strength of oceanic lithosphere?

What is the background color in fig. 4?

Do velocities in fig. 6 come with uncertainties?

Response to reviewer comments

Thank you for providing us with reviews of our manuscript. We have significantly overhauled the manuscript in line with the reviewers' comments, and believe that the manuscript is much stronger as a result of this feedback. Please find a track-changes document attached to this submission where we have indicated all alterations to the manuscript compared to the original submission. Below is our point-by-point response to the reviews in blue.

Reviewer #1 (Remarks to the Author):

This manuscript addresses a topic that has been in need of attention for quite some time. Considerable research has considered plume-ridge interaction from geochemical, geophysical, and geodynamics perspectives, but those studies have not directly addressed the specific mechanism of how plumes interact with ridges and affect their migration via anchoring/de-anchoring. The multidisciplinary approach taken by the authors that combines geodynamic modeling with geochemistry to “ground truth” the modeling is an effective one, for the most part, but not as clearly conveyed or applied as it could be in the manuscript, something easily addressed. The idea that plume flux into a ridge system is a controlling factor in terms of how effectively a ridge can be anchored to a plume is an important concept to have identified in a systematic way. In short, I anticipate that this paper will be appreciated and referenced by many researchers working on near-ridge plumes and near-plume ridges in the future.

Some comments as I go through the manuscript:

1. I am not a modeler, so I cannot assess the model details or methodology. That said, it would help readers like me who are not modelers to clarify a bit of the Results section. For instance, why was a 323 Myr timespan chosen? At the most basic level, is this modeling a generic spherical planet with Earth-like mantle convection or one that has some of Earth's characteristics (plates, plumes, etc.)? Stating this unequivocally up front would eliminate some questions to those who are not in the modeling realm, and would help make this more friendly to the non-modeler.

The 323 Myr timespan was simply an arbitrary point at which to stop the numerical simulation. All simulations need to balance model resolution with the number of timesteps, the numerical complexity of solving each timestep, and the computational resources available to run the simulation. A detailed explanation of the choices of parameters, initial conditions, and rheological scaling is available in the methods section. We have also added some text to the results section to clarify some general details of the simulation. Importantly, we write, “the model produces self-consistent plate-like behaviour and dynamic mantle plumes which reproduce plume excess temperatures, and buoyancy fluxes statistically comparable to Earth observations” (lines 73-75).

2. Figure 1 could use a bit more explanation in the caption of how the flux was modeled; just recapping it in a sentence in the caption would help make this a more understandable figure.

We have included a sentence in the Fig 1 caption explaining how plume buoyancy flux was calculated, referencing the methods section for more details.

3. Figure 2 would benefit from some explanation in the caption to indicate that it's effectively showing 3 dimensions; at first it is confusing b/c it looks like it's a bird's eye (2D) view, but I think (?) that it's meant to show the tilt in the plume conduit? If this is incorrect, then it supports my point that it's a bit hard to know how to understand this figure clearly. Same applies to the two animation clips.

We have added more shading to the plume conduits to improve the perception that it is a 3D illustration. The core-mantle boundary (CMB) is now indicated on the figure and the caption has been modified to clarify the illustrations are 3D.

4. In Results (and in the model), is the plume flux changing in response to the interactions with the ridge, or is it changing b/c it's a generic plume lifetime that includes increasing in intensity then waning flux as it goes extinct? What I'm trying to ask is whether plume flux is tied to ridge motion in some way, because around line 94, the paper reads as though the flux is being documented, not pre-determined. I am sure I've missed something here, but stating clearly (even if it's obvious to modelers) that you are or are not modeling the plume flux behavior would help. This section is written as though the plume flux is a response to the model parameters, but I'm unclear if that's the case in reality. Again, if I missed something, apologies. NOTE: I think I sorted this out, see point 21...if that's correct, some of these comments may not be as relevant; if that's NOT correct, then definitely things need to be clarified.

There is no forcing of plumes or plume buoyancy flux, rather we allow the plumes to evolve self-consistently within the simulation and track their characteristics through time (i.e. excess temperature, upwelling velocity, plume flux, etc.) Any interaction between plumes and mid-ocean ridges occurs spontaneously within the model. In line 75 we articulate this point more clearly, "From the long model timespan of 323 Myr, we track the evolution of multiple mantle plumes and spreading ridges through time. There is no forcing of plume or spreading ridge dynamics which enables us to study feedbacks in constantly evolving plume-ridge interactions."

5. Line 96: Is this meant to be Figure 2a or 3a? Seems confusing to be jumping to Figure 3 when the paper is mostly focused on Figure 2 here.

Figure 3a is correct. The results section have now been split into more paragraphs which deal with one figure or concept to improve readability.

6. Why does the spreading rate increase again (Figure 3g) once the plume has been disconnected from the ridge? This seems counter-intuitive from a physical perspective and definitely needs to be discussed.

In cases where strong asymmetric forces are exerted on a plate pair, the spreading ridge bordering the spreading rate needs to increase to accommodate ridge migration. In lines 151-153, we clarify this process: “The seafloor spreading rate increases from 2 to 8 cm/yr (Figure 3g) to accommodate rapid ridge migration following a change in plate boundary forces associated with established far-field subduction zones (Figure S2f).”

7. The Results section would be easier to follow if it were split into shorter, single-focus paragraphs, each dealing with one figure/phenomenon at a time.

We have split the results section into several smaller paragraphs to improve readability.

8. Line 113-114 how old is “older” seafloor? How much older can it be in a near-ridge plume system?

Agreed. After re-reading this paper the age of oceanic seafloor appears to be less important than spreading rate. This has been rectified.

9. Line 118: Within 1 Myr of what?

This has been rephrased to “The timescale of the ridge jump occurs within 1 Myr...” to avoid confusion.

10. Line 121: Need a “within” in front of 1000 km, and the “which” should be a “that”

Corrected.

11. Caption of Figure 4: Isn't it kind of a given that “plumes that are active over a longer timespan within the numerical model tend to have a larger buoyancy flux” or is that not the case? Would be good to clarify this statement further. Furthermore, is this a finding, or a constraint of the model?

It is intuitive that stronger plumes should outlive weaker plumes, but to my knowledge this has not been demonstrated in previous research. This has been clarified in the caption. There is no constraint in the numerical model to enforce plume longevity. As stated in the results (lines 74-76): “The model produces self-

consistent plate-like behaviour and dynamic mantle plumes which produce plume excess temperatures, and buoyancy fluxes comparable to Earth observations”.

12. Paragraph beginning line 146: This pgh can use reworking to clarify its content. The part beginning “from the 13 instances of...” conveys an important finding, but it is delivered after explaining why that might be the case, so it’s hard to follow. Also, are plumes being considered that do not get to the surface? That’s implied in line 151 with the plume reaching a “shallower depth.”

We have moved this paragraph to the top of the section entitled “Ridge de-anchoring” since it provides general statistics on de-anchoring over 13 plume-ridge interactions. From there we focus on one representative example of plume-ridge de-anchoring discussing the geodynamic drivers for this phenomenon.

Shallower depth refers to how much the plume penetrates the overriding plate. Indeed, the plume reaches the surface. The statement considers the qualities of the plume that localize deformation through thermal weakening. Increased upwelling velocity means more of the plume penetrates the overriding plate, and increased temperature weakens the overriding plate. We have reworded this part to, “A stronger buoyancy flux equates to a greater upwelling velocity and higher temperature. The former means the plume penetrates a shallower depth in the overriding plate and the latter results in lower viscosity. Both are weakening agents which maintain the localisation of deformation in the overriding plate which, when reduced, facilitates ridge migration.” (Lines 163-167).

a. And how does plume viscosity matter here; if the ridge is being anchored, it’s presumably somehow related to the mantle’s viscosity/physical properties; what is the role of the plume’s viscosity in this system?

Plume viscosity does not matter very much in terms of anchoring spreading ridges. The plume viscosity in the numerical models is relatively low, so it does not impart stresses on the overriding plate. Plumes act more through their thermal effect beneath the lithosphere, leading to its local thinning.

13. Figure 6: Should label the India-Australia ridge.

Added.

14. Line 176: What is the justification for the “most extensive plume products” mentioned here? Volume? This needs a citation at the minimum and ideally some graphic/explanation of how this is known, given how impactful this evidence could be for the argument laid out in this paper.

a. And why would a tilted plume result in more extensive plume products, wouldn’t more

extensive material result from higher fluxes, not plume angle? Or is more widespread plume materials the appropriate description (as it's described in the final paragraphs)?

We have removed this sentence on the focusing of plume products on the Indian plate from a tilted Kerguelen plume. This behavior is predicted only from the numerical models and we have no way of knowing if the Kerguelen plume was tilted at 80 Ma during the formation of the Ninety-East Ridge. Magma production, on the other hand, is focused on the Indian Plate, which is evidenced by the formation of the Ninety-East Ridge and relatively little magma production on the Australian or Antarctic plates during from 80 to 43 Ma (Figure 6).

15. Line 185: I'm not so sure that the ratio between MORB and more enriched plume compositions reflects changes in the capacity of the plume to deliver more deeply sourced melt to the surface so much as on whether the plume is diluted by entrained upper mantle and/or depleted lower mantle material. The Zindler et al. (1979) reference predates a lot of research into this topic that has happened since; this warrants clarification/revisiting/updating.

The first-order observation that spreading ridge segments on or near hotspots exhibit geochemical evidence of mixing of plume and asthenosphere has been confirmed over 50 years (since Schilling and co-workers). And bigger plumes (greater buoyancy) that are directly beneath or close to ridges have stronger and wider (along ridge) geochemical signatures than weaker or more distant plumes. Details of how plume and asthenosphere mix (e.g. entrainment, depending on viscosity difference) and whether the plume delivers multiple components from the lower mantle are interesting but secondary. Our simple assumption is that the Kerguelen plume has delivered the same composition (that of the Kerguelen Island, ~25 Ma volcanics) through time, that mixes variably with the asthenosphere (SEIR composition) depending on the buoyancy flux of the plume. The correlation we see between isotopic compositions and magma flux (plume buoyancy) is consistent with this.

a. Similarly, in the next sentence, stating that plume buoyancy flux plays a first-order control on the spread between depleted and enriched endmembers is also at the minimum an oversimplification, and really more of an inaccurate generalization.

We agree that this is a simplification, but contend that it is an accurate one that applies generally to plume-ridge geochemical signatures.

The Kerguelen plume has an EM1 mantle source composition. Simplistically, greater buoyancy flux = higher mantle temperatures, greater volumes of melting, and greater proportion of plume material in the mixing zone with ambient asthenosphere. So, the composition of the volcanic product should then be closer to the plume composition. This ignores complexities such as possible

entrainment of asthenosphere into plume, and any changes in the plume endmember composition with time, or multi-component plume compositions. However, such simple mixing is evident along spreading ridges e.g., Reykjanes Ridge [Ito *et al.* 2003] and Galapagos Spreading Center [Schilling *et al.* 2003], and the extent is correlated with plume buoyancy (as indicated by volume of melt products). Geodynamic laboratory and modeling studies predict that the width over which plumes expand along the spreading ridge axis increases with plume flux and buoyancy, and decreases with plate spreading rate, plume viscosity, and plume-ridge separation.

16. Line 190: the first $^{143}\text{Nd}/^{144}\text{Nd}$ should presumably be $^{87}\text{Sr}/^{86}\text{Sr}$?

Yes, fixed.

17. It would really clarify this isotope argument to produce a simple timeline of the different isotopic ratios in samples, instead of the x-y isotope plots. It's very hard to piece this together to understand the proposed pattern.

The distribution of isotopic compositions in bivariate plots are important to include to examine the variations between N-MORB and enriched plume end-members, but we now also include isotopic compositions vs. latitude (a proxy for age) along the Ninety-East Ridge.

a. Also, why include N Atlantic Ridge (is that meant to be Australian?), or Pitcairn if they are ancillary to the story? Simpler would be better.

We removed Pitcairn and the Australian-Antarctic Ridge fields from the isotopic composition plots (Figure 7). The fields we plot include the SE Indian Ridge (N-MORB), 24-25 Ma Kerguelen Archipelago basalts (Enriched Plume), Amsterdam-St. Paul Islands, and Kerguelen Plateau basalts. These greatly simplify the interpretation of geochemistry results.

18. The model could use clarification and simplification in how it's explained. Is the idea that when the plume is anchoring the ridge, we should see more intermediate signatures along the two-component spectrum, but more plume-like signatures when the plume is on its own/ridge-free? (which is pretty much the basis of most plume-ridge interaction models that use geochemistry as the metrics)

We say nothing about plume—ridge distance, only variable plume buoyancy flux. Between 80 – 43 Ma, the plume is coupled to the 90E ridge. The changes in radiogenic isotope is due to variations in the plume buoyancy flux adding excess material (measured by the magma production) which exhibit a more “enriched plume” signature.

a. If so, the PCA doesn't add much to this; seems like you don't need it here to test your hypothesis. PCA kind of muddies the water since the only focus is on PC1, which is just two-endmember mixing. What is involved in PC2, then, and why is it needed at all? Why not just use average plume and average MOR compositions (as is done in the binary isotope plots already)?

Thank you for this suggestion. We have removed the principal component analysis (PCA), revised the bivariate isotope analysis (Figure 7) and created new plots of Sr, Nd, Pb ratios as a function of latitude (along the 90E ridge). The PCA required that any rows with missing data be omitted from the analysis, which excluded a considerable number of samples along the 90E ridge. Dropping this in favor of bivariate plots and isotope ratios plotted against latitude increases the number of samples we can include, which bolsters our comparison between magma flux and radiogenic enrichment.

19. Again, similar to point 18 as I try to work this out: If I am understanding the proposed isotopic mixing model, we should be seeing more ridge-like compositions first along the 90E ridge's oldest samples, then a shift to more plume-like/intermediate signatures, then back again to ridge only for the youngest ones...right? This model could be conveyed much more clearly if that's what it is; if not, then it definitely needs more explanation; the reliance on PCA to get the point across isn't that effective. It would also be much more clearly conveyed in a simple graphic with time along one axis, as stated above, than forcing the reader to decipher it here by color (Figure 7).

Refer to the response above, and our response to point 21.

a. Also, the model works ok for (a) and (b) in Figure 7 (Sr, Nd, Hf isotopic ratios), but it really doesn't work well for either $^{206}\text{Pb}/^{204}\text{Pb}$ - $^{208}\text{Pb}/^{204}\text{Pb}$

This is because the isotopic compositions in (a) and (b) are more strongly correlated and dispersed. The dispersion in $^{206}\text{Pb}/^{204}\text{Pb}$ is not great so the range in MORB-plume mixing is not as clear.

(c) (why don't the younger samples get more MORB-like than they do, but stay closer to the plume composition?), and why do the older samples in the Sr-Pb system (d) not sit closer to N-MORB compositions?

The older samples (higher plume buoyancy) should be closer to the plume endmember, which they are. The younger samples get more MORB-like which is consistent with a waning plume prior to de-anchoring.

The model should work in all isotopic systems, but it doesn't from what I can see, which calls it into question and warrants attention.

b. Similarly, if this is an issue of going from ridge-only to plume/ridge mixing to ridge-

only again, why are there so many inflection points in magma flux in Figure 9(b)? What happened to cause those complexities in the magma flux vs latitude (time) curve? Same question applies for the Walvis/Tristan analysis in the supplemental docs.

Refer to point 21.

20. Line 203: What's "magma flux" referencing here? In the plume or the ridge, and how does it differ from buoyancy flux? I assume it's MOR flux?

Refer to point 21.

21. Ok so maybe the issue I'm having is that the authors mean buoyancy flux/plume contribution to the ridge whenever it is mentioned, not in fact changing plume flux over time? If so, it took me until nearly the end of the paper to understand that nuance, despite careful reading. Some simple clarification early on to indicate that the study focuses on the RIDGE's erupted products and the buoyancy flux of magma to the ridge (from both the conventional MOR upwelling plus or minus plume contributions) would probably sort out a lot of the potential for confusion. I'm sure it's mentioned early on, but as someone who works on plumes, I definitely assumed that "buoyancy flux" referenced the PLUME's buoyancy flux, not the flux of material to the MOR throughout this paper. My error if I missed it early on, but many people read papers far more quickly, so finding a way to make it bomb-proof what you are modeling early on in the paper would be great (just bear in mind that the use of "buoyancy flux" is most commonly and automatically attributed to the PLUME among plume researchers, not the supply to the ridge).

Yes, we mean plume buoyancy flux which adds to the normal asthenosphere melting beneath the spreading ridge, resulting in increased magma production (e.g. the Ninety-East Ridge). In detail, much of the combined melting will erupt at the ridge, but there will be additional volcanic products adjacent to the ridge (e.g., Iceland).

In Nature, plume buoyancy flux is difficult to constrain at times older than the present-day. It is this which motivates our analysis of magma production and geochemistry along the Ninety-East Ridge so that we may better understand the relative 'strength' of the Kerguelen plume through time. In particular, we explore whether the Kerguelen plume was waning in the leadup to plume-ridge de-anchoring at 43 Ma, which we have demonstrated.

The geochemistry of the Ninety-East Ridge has been moved to follow the section on magma production and we have removed the PCA analysis. This greatly improves the flow of arguments and clarifies the important point that the input of

the Kerguelen plume adds excess, fusible material beneath the SE Indian ridge, which is manifested in the production of the Ninety-East Ridge from 80 to 43 Ma.

22. Line 246: Really can't follow what's meant here as written. Could use clarification.

“Interaction” should be “attraction”. This has been fixed.

23. Are there any implications of this model for the ocean islands and their compositions (and/or morphologies/sizes) being formed by the plume in a near-ridge environment?

We have added the following sentence in line 236 “Following de-anchoring, the volcanic products should switch from MORB to OIB systematics as intraplate volcanism is established.”

24. In general, the manuscript would benefit from a detailed proofread that focuses on clarifying sentences, finding a more systematic way to convey information via figures (it bounces around between figures quite a bit, which makes it frustrating to read), designing shorter paragraphs focused on single topics, and use of commas and other minor grammatical tools to help clarify sentences (e.g., line 146 needs a comma after “anchor” to make it read easily).

We have split large paragraphs in the results section into several smaller paragraphs that deal with one concept and one figure at a time to improve readability. The manuscript has also been thoroughly checked for punctuation.

Reviewer #2 (Remarks to the Author):

I have reviewed the manuscript 'Spreading ridge migration enabled by plume-ridge de-anchoring', by B. Mather and coauthors. This manuscript uses numerical models of viscous flow to draw conclusions about the dynamics of ridge migrations and the controls operated by ridge-plume interactions.

The topic of the study is one of interest and has the potential to advance tectonic reconstructions at the global scale. However, I would see this study more suited for a specialized outlet. In fact, the study revolves around analyses of model behavior upon changes of the input parameters. I fail to see a testable hypothesis that is rooted in specific observations, and that is tested against independent data and uncertainties. The study makes a comparison with SE Indian Ridge reconstructions, but to me this seems more a case of resemblance rather than prediction. This is not to say there is no merit in this study. Rather, that it does not make a convincing case for a paradigm shift in geological knowledge.

Our study explores how plume-ridge interaction can influence plate boundary migration through time. We show that the interactions between plumes and

ridges are modulated by plume buoyancy flux which can anchor plate boundaries in-place. We relate the findings from numerical models to examples from nature by examining the magmatic products of plume-ridge interactions. By analysing magma production and radiogenic isotope variation along the Ninety East Ridge, we reconstructed the changes in plume buoyancy flux of the Kerguelen plume to the SE Indian Ridge. This adds support to our models of plume-ridge interactions which show that plumes of relatively low buoyancy flux are statistically correlated to spreading ridge migration. These predictions also hold for the Walvis Ridge in the South Atlantic (Figure S4). Previous studies (mostly 2D) have examined plume-ridge interaction in a closed system, but our study is the first to examine self-consistent plumes and their interactions with constantly evolving plate boundaries. This has important implications for the evolution of plate boundaries, the distribution of plume-related volcanism, and the reconstruction of ocean basins through deep time.

Below are a few points that authors may find useful:

That plate tectonics is primary driven by slab pull is a rather partial view: there is a significant body of work illustrating the relevance of controls on plate tectonics operated by pressure-driven mantle convection (for instance, the work of Lenardic, Richards and coauthors). Some of the statements that are key to the dynamics proposed in this study are based on other studies (ref. 2 for plume push, ref. 11 for slab pull) that contain proposals rather than tested hypotheses. This gives the impression that these are well established dynamics, which is not the case.

It is well accepted in the tectonics and geodynamics communities that slab pull and (to a lesser extent) plume push are primary drivers of plate motion. The references we have cited (ref. 1 for slab pull, ref. 2 for plume push) provide compelling evidence for their respective roles in driving plate motion, which has been embraced by the scientific community as reflected by the high number of citations. More recently, pressure-derived mantle convection (Lenardic), similar to mantle drag (Coltice), has been proposed as additional driving forces of plate motion, although the magnitude of this force is poorly understood.

The mechanism proposed by Lenardic, Richards, and co-authors does not contradict the evidence that slab pull and plume push forces drive plate motion. Rather, they propose that asthenosphere flow velocities do not always align with the direction and magnitude of plate motion. This line of questioning is tangential to the aims and outcomes of our study. We propose that strong plumes act as an anchor thereby impeding ridge migration. Whether plate motion is driven by slab pull or pressure-driven mantle convection is of minor importance.

A notion invoked at line 105 is that a strong plume can break the lithosphere. This statement should be backed up with either references (the past work of Roger Buck

may be something to look at) or tests/calculations. For instance, can the push of a plume exceed the strength of oceanic lithosphere?

It is not the case that plumes push through oceanic lithosphere in the geodynamic models we use. Rather, plumes are agents of thermal weakening. As explained in lines 163-167: “A stronger buoyancy flux equates to a greater upwelling velocity and higher temperature. The former means the plume penetrates a shallower depth in the overriding plate and the latter results in lower viscosity. Both are weakening agents which maintain the localisation of deformation in the overriding plate which, when reduced, facilitates ridge migration.”

What is the background color in fig. 4?

The blue color is a kernel density estimate based on the density of points. We have updated the caption accordingly.

Do velocities in fig. 6 come with uncertainties?

Unfortunately, there are no uncertainties associated with this plate model due to the qualitative nature of interrogating palaeomagnetic data to reconstruct plate motions. However, since we reconstruct plate motions only back to 60 Ma, the velocities are relatively well constrained by the seafloor magnetic anomalies which are preserved in seafloor of the Indian Ocean.

REVIEWERS' COMMENTS

Reviewer #1 (Remarks to the Author):

The manuscript is significantly clearer and more impactful in this revised version. It reads smoothly and makes an admittedly somewhat circumstantial but intriguing case for the role of plumes in not only anchoring MORs but also in how de-anchoring from a plume can affect ridge migration rates to a significant extent. Some of the claims are not that well supported, particularly near the end of the manuscript where it gets especially speculative about the role of de-anchoring, plume force balancing, and increases (or not) in MOR migration rates. Still, these are interesting ideas that should be explored, so they seem appropriate to include in the paper. That said, finding just a bit more (even circumstantial) evidence to support some of these claims (such as more locations than just Tristan to support the plate forces balancing contention and its role in MOR migration rates once a ridge has been released from a plume's anchor) would strengthen the ending of the paper.

Overall, this manuscript lays out important ideas that connect MOR migration rates, plate tectonic dynamics, and plume-ridge interaction in thoughtful and insightful ways that are likely to be both referenced and used as ideas for future work in the field.

Some minor comments:

1. Helpful intro to the Results section. The first paragraph in the Results section reads more like it belongs in the end of the introduction, in some ways, as it sets the trajectory for the work in the paper.
2. Super minor suggestion: use "that" when there is no comma in front of it, or add a comma in front of "which" when used. Not like this affects the science...9
3. I would be helpful to mention somewhere the range of fluxes of existing mantle plumes (the authors do for Hawaii, but no others), just for context for those unfamiliar with plume flux values.
4. Figure 7: Why are there two decreases in enrichment level along the plume track? This model predicts enriched to more depleted to more enriched only, not the additional enriched/depleted cycle. It would be great to hear the authors' perspectives on this additional cycle in the context of their proposed model.
5. Small (1995) should really be cited (about ridge jumps owing to thermal influence of the plume on a ridge), given that it's one of the earliest works that invoke this kind of model.
6. Line 266: opening sentence could use clarification/refinement. Verb/subject agreement isn't right, and "similar" needs definition (similar to each other, presumably). Also, what does "much" mean in "much more slowly?" And more slowly than when anchored, presumably? This could all use some qualification to make the use of the adjectives more precise. Minor issue that would clarify.
7. Line 268: Again, "similar" refers to what exactly? Should be defined to be unambiguous. This whole paragraph is pretty speculative and poorly constrained (both as written and conceptually), and could use more precise language as well as referencing to support claims made in it.
8. No hyphen for "in-place" in line 276.
9. In general, the last few paragraphs get a bit too speculative and poorly constrained, and could benefit from reworking and stronger grounding in the scientific literature, to at least ground it in some corroborating evidence to support the claims (even if it's circumstantial, more than is included here

would help).

Review Mather et al. 2024, Nature Communications

In this paper, the authors combine the results from a numerical model of mantle convection with geochemical data and insights from plate tectonic reconstructions to propose and quantify a new mechanism of plume-ridge interaction from ridge capture to de-anchoring. I really like that the authors used a range of methods and consider real-world implications of their work with a case study. I really enjoyed reading this paper! It's really cool and overall it was quite clear!

Obviously, I did not see the first version of this manuscript and for some reason I couldn't find the tracked changes file, but from reading the rebuttal letter, I gather that the authors made quite significant changes to the manuscript. I think that - with a few exceptions - most comments from the previous reviewers have been addressed satisfactorily. The manuscript reads well even though it is quite dense (a lot of information and findings in so few words!). The figures are really clear and beautiful. All in all, I think the paper is in quite a good shape.

I still have a few comments about the results and implications of the study (see below) that perhaps require some additional paragraphs and additions to figures plus a bunch of minor textual comments for some last clarifications to the text. As such I recommend moderate revisions before this manuscript is ready for publication.

PS: I received 2 versions of the manuscript: one that included the methods and one that didn't. As a result, the line numbers are inconsistent in the latter parts of those two versions of the manuscript. So, when I refer to line numbers, I note the section of the paper as well, so you can hopefully find the line I am talking about across the different versions.

Major comments

1. As far as I can discern, there is only 1 model used in this paper: model 2 from Arnould et al. 2020, which is then thoroughly analysed to suss out the details of plume-ridge interactions. Is that correct? Because at multiple instances in the paper, the term 'numerical models' is mentioned. For example, the section title in the methods "setup of whole mantle convection models" and line 388 in the caption of figure 1: "the numerical models". This could lead the reader to believe that there are in fact multiple models, while that is not the case. So, there needs to be a thorough search throughout the manuscript to make sure the text only ever mentions 1 numerical model. As a further example that this can be easily misunderstood, I think this is something reviewer 2 picked up on, who references "analyses of model behaviour upon changes of the input parameters", but as far as I can see, there are no input changes / parameter studies at all throughout this work. This is fine, but best to be careful with the formulation in the manuscript then.
2. It would be useful and interesting to show the difference in plume buoyancy flux between ridge capture and ridge release for the 13 plume-ridge interactions in another boxplot in Figure 1 to show that the plume buoyancy flux systematically decreases (and by how much) between ridge capture and ridge release, because that info is not currently not available in the plot, but would really corroborate the narrative of your manuscript, right?

3. Why do the plume buoyancy fluxes increase / decrease depending on their interaction with the ridge? Is that just the convection pattern? Or is there some other mechanism at play? What controls the plume buoyancy flux? It would be great if you could discuss that briefly.

In line with points 2 and 3, I guess what I am hinting at is that I think there is a kind of chicken-and-egg problem here: does the plume buoyancy flux decrease and therefore the ridge is free to de-anchor and move away? Or does the ridge migrate away and does this cause a reduction in plume buoyancy flux? I know you argue for the former explanation in this article and indeed the right panels of Figure 3 seem to support your conclusion, but this was only 1 of the 13 interactions between plumes and ridges and hence - presumably - 1 of 13 examples of ridge de-anchoring. So, is the explanation you propose always what you see in the model? Can you really reject the other hypothesis of the ridge moving away first and that causing a reduction in plume buoyancy flux? It would be great if you could add some discussion on this and show the results of the other (12) examples of ridge capture and ridge de-anchoring in the supplementary material for reference (analogous to Figure 3 perhaps).

4. Since triple junctions seem to be formed during ridge capture and when the plume and ridge are coupled, I just wonder if this implies that these plume-ridge interactions are one of the main methods of forming a triple junction? Are all current triple junctions on the Earth fed by / on top of a plume for instance? And does this mean that triple junctions are by definition a temporary, transitional feature that only form through ridge capture? Do you see the formation of triple junctions for all of your 13 plume-ridge interactions? How long-lived are they, i.e., do they last for the entire duration of the ridge-plume coupling? These are many questions - I apologise - but they struck me while I was reading your paper, so it would be really cool if you could expand a bit on this / provide some discussion in the manuscript about this.
5. I agree with reviewer 1 that Figure 8b and c and to an extent d (showing eNd against eHf ; $^{208}Pb/^{204}Pb$ against $^{206}Pb/^{204}Pb$; and $^{204}Pb/^{204}Pb$ against $^{87}Sr/^{86}Sr$) are not convincing in showing that younger samples have a more N-MORB composition compared to older samples and hence do not convincingly corroborate the main idea of the paper. It might be very convincing to geochemists, but as a geodynamicist/seismologist, I was a bit underwhelmed. I know the authors wrote an explanation to the question of reviewer 1, explaining why it is expected that these isotope variations do not necessarily show this trend (because of less dispersion in the samples...? what does that mean exactly? and why is that?) but I did not pick up on this while reading the manuscript, so please make this more clear in the main text.

Moderate comments

1. If I understand correctly, the ridge capture and ridge de-anchoring examples in Figures 2, 3, 5, S1 and across the text are actually from 2 different plume-ridge interactions. That is, 2 of the total of 13 mentioned in the text. It took me a while to figure this out and as a result I was initially quite confused why the left panels of Figure 3 don't seamlessly transition into the right panels of Figure 3. Could you make this more clear in the manuscript?
2. In the caption of Figure 1, you mention plume-ridge coupling. Am I to understand that that means "anchoring"? That is, the ridge being on top of the plume? Could you clarify this and/or provide a definition for plume-ridge coupling?

Minor comments

1. Line 74 in the results: I agree with reviewer 1 that the model time of 323 Myr is oddly specific and invokes the idea that there might be some scientific relevance to this number. Perhaps adding the word "arbitrary" in front of "model timespan" would resolve this issue.
2. Line 81 in the results: Should this be "*within* 1000 km of a ridge"?
3. Line 319 in the methods: "by from" → remove "by"
4. Line 320 in the methods: "encircle" → "encircles"
5. Line 378 in the code availability: "is freely available" → "are freely available"
6. Line 379 in the code availability: "ET" → "ETH"
7. Line 397 in the caption of Figure 1: "begin" → "begins"
8. Caption Figure 2: Define the abbreviation "MOR" (and maybe also "MORB" in the text where you first use it. I know you write "spreading ridge basalts (MORB)", but I always thought it stood for "mid-oceanic ridge basalts". Would be good to clarify the origin of this abbreviation in the manuscript.
9. Caption Figure 3: Could you make clear that there are 2 different y-axes in the left and right panels? Because it is very easy to assume at first glance that there are the same and it makes the comparison a bit difficult. Would it be possible to have the same y-axes for the panels? or maybe have that figure for comparison in the supplementary material?
10. Line 430-431 in caption of Figure 3: It seems like this sentence should be moved down a bit. Perhaps in the explanation of panel d.
11. Caption Figure 5: Please make it clear in the caption that the x-axes between the top and bottom panels differ, because one naturally wants to compare the two, but that actually doesn't really work.

Response to reviewer comments

We thank the editor and the reviewers for providing feedback on our manuscript. We believe this is a significant improvement on our previous version and thank the reviewers for their helpful suggestions. Our responses to the reviewers' comments below are highlighted in blue.

Reviewer #1

The manuscript is significantly clearer and more impactful in this revised version. It reads smoothly and makes an admittedly somewhat circumstantial but intriguing case for the role of plumes in not only anchoring MORs but also in how de-anchoring from a plume can affect ridge migration rates to a significant extent. Some of the claims are not that well supported, particularly near the end of the manuscript where it gets especially speculative about the role of de-anchoring, plume force balancing, and increases (or not) in MOR migration rates. Still, these are interesting ideas that should be explored, so they seem appropriate to include in the paper. That said, finding just a bit more (even circumstantial) evidence to support some of these claims (such as more locations than just Tristan to support the plate forces balancing contention and its role in MOR migration rates once a ridge has been released from a plume's anchor) would strengthen the ending of the paper.

Overall, this manuscript lays out important ideas that connect MOR migration rates, plate tectonic dynamics, and plume-ridge interaction in thoughtful and insightful ways that are likely to be both referenced and used as ideas for future work in the field.

Some minor comments:

1. Helpful intro to the Results section. The first paragraph in the Results section reads more like it belongs in the end of the introduction, in some ways, as it sets the trajectory for the work in the paper.

The first paragraph of the results section introduces the numerical model which frames the beginning of the results section, so we think it is best to keep it where it is. Explaining the numerical model in detail does not seem relevant to the introduction. In the introduction we explain the motivation for this study and frame the ideas we explore later in the paper.

2. Super minor suggestion: use "that" when there is no comma in front of it, or add a comma in front of "which" when used. Not like this affects the science...

Fixed.

3. It would be helpful to mention somewhere the range of fluxes of existing mantle plumes (the authors do for Hawaii, but no others), just for context for those unfamiliar with plume flux values.

Good point. We now provide an upper and lower range of plume fluxes. The upper end being Hawaii (2,800 – 8,700 kg/s) and the lower end being Galapagos (1,000 – 1,800 kg/s) on lines 123-125.

4. Figure 7: Why are there two decreases in enrichment level along the plume track? This model predicts enriched to more depleted to more enriched only, not the additional enriched/depleted cycle. It would be great to hear the authors' perspectives on this additional cycle in the context of their proposed model.

Plumes are dynamic and will exhibit varying buoyancy fluxes throughout their evolution. We attribute these local decreases in enrichment along the plume track primarily to a small reduction in plume buoyancy flux. Additional sources of variation may occur due to the entrainment of upper mantle into the upwelling plume or variations in plume source composition with time. We have added this in the text (222-223).

5. Small (1995) should really be cited (about ridge jumps owing to thermal influence of the plume on a ridge), given that it's one of the earliest works that invoke this kind of model.

We have included a reference to Small, 1995.

6. Line 266: opening sentence could use clarification/refinement. Verb/subject agreement isn't right, and "similar" needs definition (similar to each other, presumably). Also, what does "much" mean in "much more slowly?" And more slowly than when anchored, presumably? This could all use some qualification to make the use of the adjectives more precise. Minor issue that would clarify.

We have revised this sentence for clarity. It now reads, “If the balance of forces on each plate pair are similar to each other, then the spreading ridge would migrate much more slowly following plume-ridge de-anchoring than if the plate force balance was highly asymmetric.” (Lines 282-284)

7. Line 268: Again, “similar” refers to what exactly? Should be defined to be unambiguous. This whole paragraph is pretty speculative and poorly constrained (both as written and conceptually), and could use more precise language as well as referencing to support claims made in it.

We have revised this sentence for clarity. It now reads, “The Atlantic Ocean is not bounded by subduction zones which indicates the balance of plate boundary forces along the Atlantic Ridge is relatively similar to each other²⁸.” (Lines 284-285). We have added a reference to Clennett *et al* 2023 (ref 28) on assessing tectonic plate driving forces.

8. No hyphen for “in-place” in line 276.

Removed.

9. In general, the last few paragraphs get a bit too speculative and poorly constrained, and could benefit from reworking and stronger grounding in the scientific literature, to at least ground it in some corroborating evidence to support the claims (even if it’s circumstantial, more than is included here would help).

We have expanded the discussion around how plume-ridge de-anchoring facilitates ridge migration depending on the force balance on each plate pair. We have added a reference to Clennett *et al* 2023 (ref 28) on assessing tectonic plate driving forces. A new example is provided in the Easter plume – East Pacific Rise de-anchoring, now discussed on lines 293-298.

Reviewer #3

In this paper, the authors combine the results from a numerical model of mantle convection with geochemical data and insights from plate tectonic reconstructions to propose and quantify a new mechanism of plume-ridge interaction from ridge capture to de-anchoring. I really like that the authors used a range of methods and consider real-world implications of their work with a case study. I really enjoyed reading this paper! It’s really cool and overall it was quite clear!

Obviously, I did not see the first version of this manuscript and for some reason I couldn’t find the tracked changes file, but from reading the rebuttal letter, I gather that the authors made quite significant changes to the manuscript. I think that - with a few exceptions - most comments from the previous reviewers have been addressed satisfactorily. The manuscript reads well even though it is quite dense (a lot of information and findings in so few words!). The figures are really clear and beautiful. All in all, I think the paper is in quite a good shape.

I still have a few comments about the results and implications of the study (see below) that perhaps require some additional paragraphs and additions to figures plus a bunch of minor textual comments for some last clarifications to the text. As such I recommend moderate revisions before this manuscript is ready for publication.

PS: I received 2 versions of the manuscript: one that included the methods and one that didn’t. As a result, the line numbers are inconsistent in the latter parts of those two versions of the manuscript. So, when I refer to line numbers, I note the section of the paper as well, so you can hopefully find the line I am talking about across the different versions.

Major comments

1. As far as I can discern, there is only 1 model used in this paper: model 2 from Arnould *et al.* 2020, which is then thoroughly analysed to suss out the details of plume-ridge interactions. Is that correct? Because at multiple instances in the paper, the term ‘numerical models’ is mentioned. For example, the section title in the methods “setup of whole mantle convection models” and line 388 in the caption of figure 1: “the numerical models”. This could lead the reader to believe that there are in fact multiple models, while that is not the case. So, there needs to be a thorough search throughout the manuscript to make sure the text only ever mentions 1 numerical model. As a further example that this can be easily misunderstood, I think this is something reviewer 2 picked up on, who references “analyses of model behaviour upon changes of the input parameters”, but as far as I can see, there are no input changes / parameter studies at all throughout this work. This is fine, but best to be careful with the formulation in the manuscript then.

Indeed, we use just one model corresponding to model 2 from Arnould et al. 2020. We interrogate that model to extract multiple instances of plume-ridge interaction. The text has now been modified to remove plurals to avoid confusion.

2. It would be useful and interesting to show the difference in plume buoyancy flux between ridge capture and ridge release for the 13 plume-ridge interactions in another boxplot in Figure 1 to show that the plume buoyancy flux systematically decreases (and by how much) between ridge capture and ridge release, because that info is not currently not available in the plot, but would really corroborate the narrative of your manuscript, right?

We agree and have included another boxplot in figure 1 to show this. The caption has been updated to reflect the change. Thank you for the recommendation.

3. Why do the plume buoyancy fluxes increase / decrease depending on their interaction with the ridge? Is that just the convection pattern? Or is there some other mechanism at play? What controls the plume buoyancy flux? It would be great if you could discuss that briefly.

We measure the plume buoyancy flux over the entire depth range of the plume conduit (line 97-98). The buoyancy flux we refer to in Figure 1 and 4 is the *mean* buoyancy flux from the core-mantle boundary to the surface. While the buoyancy flux increases in the upper mantle upon ridge capture, because the upwelling velocity increases in response to spreading (Figure 5d), our view is that changes in wholesale plume buoyancy flux affects spreading ridges, not the other way around. We provide an extended discussion on this point in the response to the comment below.

In line with points 2 and 3, I guess what I am hinting at is that I think there is a kind of chicken- and-egg problem here: does the plume buoyancy flux decrease and therefore the ridge is free to de-anchor and move away? Or does the ridge migrate away and does this cause a reduction in plume buoyancy flux? I know you argue for the former explanation in this article and indeed the right panels of Figure 3 seem to support your conclusion, but this was only 1 of the 13 interactions between plumes and ridges and hence - presumably - 1 of 13 examples of ridge de-anchoring. So, is the explanation you propose always what you see in the model? Can you really reject the other hypothesis of the ridge moving away first and that causing a reduction in plume buoyancy flux? It would be great if you could add some discussion on this and show the results of the other (12) examples of ridge capture and ridge de-anchoring in the supplementary material for reference (analogous to Figure 3 perhaps).

Indeed, we argue that the plume buoyancy flux decreases which allows the ridge de-anchor and migrate away from the plume. We reject the alternate hypothesis because in all of our 13 instances of plume-ridge interaction, the buoyancy flux we refer to is the *mean* buoyancy flux from the core-mantle boundary to the surface (Figures 1 and 4). This is important because if the ridge were to influence the buoyancy flux of the plume, it would need to perturb the buoyancy flux across the entire depth range of the plume conduit in the mantle. We acknowledge that the upwelling velocity of a plume undergoing de-anchoring decreases in the uppermost mantle, which brings down the plume flux. But this would not reduce the mean plume flux significantly unless the buoyancy flux across the rest of the plume conduit was also reduced (an example of this is shown in Figure 5g). Another line of evidence is that plume-ridge de-anchoring tends to occur *after* the plume buoyancy flux has substantially decreased. For instance, in Figure 1 the boxplot for ridge release shows a reduction in mean buoyancy flux within 20 Myr window of de-anchoring, and the right pane of Figure 3 shows that de-anchoring only occurs when the buoyancy flux is already at a minimum. Therefore, we conclude it is the wholesale reduction of plume buoyancy flux which allows the ridge to de-anchor, not the other way around.

We have added additional text to the discussion to make this distinction clearer, “We contend that a reduction in plume buoyancy flux facilitates ridge migration and reject the counter argument that the migration of a spreading ridge results in a reduction in plume buoyancy flux because (1) ridge migration occurs only after the plume buoyancy flux has substantially decreased, and (2) the plume buoyancy flux is reduced from the core-mantle boundary to the surface at the time of de-anchoring. A local decrease in buoyancy flux within the upper mantle would be anticipated following plume-ridge de-anchoring, but this would not significantly reduce the plume buoyancy flux across the entire plume conduit. We observe a wholesale reduction in the mean buoyancy flux in the leadup to ridge migration (Figure 1, 5h), which suggests that waning plumes facilitate plume-ridge de-anchoring.” (lines 171-179).

4. Since triple junctions seem to be formed during ridge capture and when the plume and ridge are coupled, I just wonder if this implies that these plume-ridge interactions are one of the main methods of forming a triple junction? Are all current triple junctions on the Earth fed by / on top of a plume for instance? And does this mean that triple junctions are by definition a temporary, transitional feature that

only form through ridge capture? Do you see the formation of triple junctions for all of your 13 plume-ridge interactions? How long-lived are they, i.e., do they last for the entire duration of the ridge-plume coupling? These are many questions - I apologise - but they struck me while I was reading your paper, so it would be really cool if you could expand a bit on this / provide some discussion in the manuscript about this.

While we do observe many triple junctions being fed by mantle plumes of high buoyancy flux in the numerical models, we do not venture so far as to say that every triple junction must have been formed by a mantle plume, since our numerical models cannot capture the full complexity of plate boundary reorganisation that occurs on Earth. In the text we speculate on the association between high plume buoyancy fluxes triple junctions as follows, "High plume buoyancy flux may also be a key driver of triple junction formation, since a faster upwelling velocity and excess temperature will fragment the overriding plate into smaller plates to accommodate more efficient plate motion." (Lines 271-273). We thank the reviewer for their suggestion.

5. I agree with reviewer 1 that Figure 8b and c and to an extent d (showing eNd against eHf; 208Pb/204Pb against 206Pb/204Pb; and 204Pb/204Pb against 87Sr/86Sr) are not convincing in showing that younger samples have a more N-MORB composition compared to older samples and hence do not convincingly corroborate the main idea of the paper. It might be very convincing to geochemists, but as a geodynamicist/seismologist, I was a bit underwhelmed. I know the authors wrote an explanation to the question of reviewer 1, explaining why it is expected that these isotope variations do not necessarily show this trend (because of less dispersion in the samples...? what does that mean exactly? and why is that?) but I did not pick up on this while reading the manuscript, so please make this more clear in the main text.

The isotopic enrichment is consistent with the variation in plume buoyancy, although plume melt proportions may not be the exclusive control on isotopic compositions (i.e., entrainment of upper mantle into upwelling plume mantle, variations in plume source composition with time). The Sr, Nd isotopes show a strong correlation, revealing mixing between N-MORB and enriched plume end-members, whereas the Pb isotopes exhibit a narrower range in values, so less powerful in tracking changes in plume melt proportions. We have shown all isotopic domains to be transparent with the data and complete in our analysis. Geochemists will not be surprised by Figure 8.

Moderate comments

1. If I understand correctly, the ridge capture and ridge de-anchoring examples in Figures 2, 3, 5, S1 and across the text are actually from 2 different plume-ridge interactions. That is, 2 of the total of 13 mentioned in the text. It took me a while to figure this out and as a result I was initially quite confused why the left panels of Figure 3 don't seamlessly transition into the right panels of Figure 3. Could you make this more clear in the manuscript?

We have modified the captions to make clear that we interrogate two different plume-ridge interactions.

2. In the caption of Figure 1, you mention plume-ridge coupling. Am I to understand that that means "anchoring"? That is, the ridge being on top of the plume? Could you clarify this and/ or provide a definition for plume-ridge coupling?

The caption has been modified from "coupling" to "anchoring". Essentially, they are the same, but we make the point in the paper that plumes act as an anchor thereby impeding ridge migration (lines 161-162), thus we adopt the terminology "anchoring" and "de-anchoring".

Minor comments

1. Line 74 in the results: I agree with reviewer 1 that the model time of 323 Myr is oddly specific and invokes the idea that there might be some scientific relevance to this number. Perhaps adding the word "arbitrary" in front of "model timespan" would resolve this issue.
Done
2. Line 81 in the results: Should this be "within 1000 km of a ridge"?
Fixed.
3. Line 319 in the methods: "by from" → remove "by"
Fixed.
4. Line 320 in the methods: "encircle" → "encircles"

Fixed.

5. Line 378 in the code availability: "is freely available" → "are freely available"
Fixed.
6. Line 379 in the code availability: "ET" → "ETH"
Fixed.
7. Line 397 in the caption of Figure 1: "begin" → "begins"
Fixed.
8. Caption Figure 2: Define the abbreviation "MOR" (and maybe also "MORB" in the text where you first use it. I know you write "spreading ridge basalts (MORB)", but I always thought it stood for "mid-oceanic ridge basalts". Would be good to clarify the origin of this abbreviation in the manuscript.
Fixed.
9. Caption Figure 3: Could you make clear that there are 2 different y-axes in the left and right panels? Because it is very easy to assume at first glance that there are the same and it makes the comparison a bit difficult. Would it be possible to have the same y-axes for the panels? or maybe have that figure for comparison in the supplementary material?
We do not believe the change in scale is unclear. The y-axes have legible numbers that are on different scales. Nevertheless, we have added a note in the figure caption that points out to the reader that the y-axes have different scales.
10. Line 430-431 in caption of Figure 3: It seems like this sentence should be moved down a bit. Perhaps in the explanation of panel d.
Fixed.
11. Caption Figure 5: Please make it clear in the caption that the x-axes between the top and bottom panels differ, because one naturally wants to compare the two, but that actually doesn't really work.
We have now indicated the x-axes have different scales in the figure caption.